# MCAK recognizes the nucleotide-dependent feature at growing microtubule ends

**Wei Chen\*, Yin-Long Song, Jian-Feng He, Xin Liang\***

IDG/McGovern Institute for Brain Research, State Key Laboratory of Complex, Severe, and Rare Diseases, School of Life Sciences, Tsinghua University, Beijing, China

## eLife Assessment

This work presents **valuable** new information on the microtubule-binding mode of the microtubule kinesin-13, MCAK, the authors use quantitative single-molecule studies to propose that MCAK preferentially binds to a GDP-Pi-tubulin portion of the microtubule end. However, the evidence provided to support this claim remains **incomplete** and would benefit from more rigorous methodology particularly the diffraction limited experiments do not provide sufficient spatial resolution to support the authors' conclusions. In addition, a more through discussion of the existing literature would further strengthen the manuscript.

**\*For correspondence:**
chenwill@mail.tsinghua.edu.cn
(WC);
xinliang@tsinghua.edu.cn (XL)

**Competing interest:** The authors declare that no competing interests exist.

**Abstract** The growing plus-end is a key regulatory site for microtubule dynamics. MCAK (mitotic centromere-associated kinesin), a microtubule depolymerizing kinesin, is an end-binding regulator of catastrophe frequency. It is intriguing how MCAK specifically binds to growing microtubule ends. Here, we measure the end-binding kinetics of MCAK using single-molecule imaging and reveal that MCAK not only binds to the distalmost ends, but also to the proximal region of GTP cap where EB1 preferentially binds. Further analysis shows that MCAK strongly binds to GTPγS microtubules which mimic the GDP·Pi-tubulin-enriched region of GTP cap, and this binding preference is dependent on the nucleotide state of MCAK. This finding suggests that MCAK recognizes the nucleotide-dependent feature of microtubule ends. Moreover, we show that although MCAK and XMAP215 partly share binding regions at the distalmost ends, they act largely independently, influencing catastrophe frequency and growth rate, respectively. Overall, our findings provide new insights into how MCAK regulates microtubule end dynamics.

## Introduction

Microtubules are essential components of the eukaryotic cytoskeleton, supporting a wide range of cellular processes, including cell division, polarization, and migration (*Alberts, 2015*). However, their properties can vary considerably across different cellular contexts and structures (*Janke and Magiera, 2020*; *Kapitein and Hoogenraad, 2015*; *Wittmann et al., 2001*). This variability reflects the complex and tightly regulated mechanisms that govern the dynamics of cellular microtubules, among which the microtubule-associated proteins (MAPs) play crucial roles.

Kinesin superfamily members not only facilitate the transport of cargo along microtubules but also serve as regulators of microtubule dynamics, as exemplified by the kinesin-13 family. Initially identified through studies of spindle functions and neurons (*Aizawa et al., 1992*; *Noda et al., 1995*; *Walczak et al., 1996*; *Wordeman and Mitchison, 1995*), kinesin-13 has since been implicated in a broad range of cellular processes, including spindle assembly (*Ohi et al., 2003*; *Walczak et al., 2002*; *Walczak*

*et al., 1996*), chromosome segregation (*Kline-Smith et al., 2004*; *Maney et al., 1998*; *Rogers et al., 2004*), directional migration (*Zong et al., 2021*), ciliary length control (*Kobayashi et al., 2011*; *Piao et al., 2009*; *Vasudevan et al., 2015*), and neuronal polarization (*Ghosh-Roy et al., 2012*; *Homma et al., 2003*; *Puri et al., 2021*). The loss of kinesin-13 often results in altered structural and dynamic properties of cellular microtubules. Moreover, the subcellular localizations of kinesin-13, such as at growing microtubule ends, centrosomes, and centromeres (*Kline-Smith et al., 2004*; *Kline-Smith and Walczak, 2002*; *Mennella et al., 2005*; *Moore and Wordeman, 2004*; *Moore et al., 2005*; *Walczak et al., 1996*; *Wordeman and Mitchison, 1995*), are key regulatory sites for microtubule dynamics. These observations suggest that kinesin-13 family members are key regulators for the length, mass, and polarity of cellular microtubules.

As a representative member of the kinesin-13 family, MCAK has been characterized as both a microtubule depolymerase and a catastrophe factor (*Desai et al., 1999*; *Gardner et al., 2011*; *Hunter et al., 2003*; *Kline-Smith and Walczak, 2002*; *Maney et al., 1998*; *Walczak et al., 1996*). It has been proposed that MCAK targets growing microtubule ends by interacting with end-binding proteins (EBs) or motors, such as Kif18b (*Montenegro Gouveia et al., 2010*; *Honnappa et al., 2009*; *Lee et al., 2008*; *McHugh and Welburn, 2023*; *Tanenbaum et al., 2011*). Upon binding, MCAK destabilizes the protective GTP cap at growing microtubule ends by bending and disassembling underlying substrate, including terminal tubulin dimers or protofilaments (*Benoit et al., 2018*; *Moores and Milligan, 2006*; *Wang et al., 2017*), thereby increasing the frequency of catastrophe events. MCAK can also function independently (*Desai et al., 1999*; *Gardner et al., 2011*; *Hunter et al., 2003*), potentially targeting stabilized microtubule ends through direct binding or rapid diffusion along the microtubule lattice (*Desai et al., 1999*; *Helenius et al., 2006*; *Oguchi et al., 2011*; *Wang et al., 2012*). Previous structural studies suggest that MCAK recognizes curved protofilaments at microtubule ends (*Asenjo et al., 2013*; *Benoit et al., 2018*; *Tan et al., 2008*; *Trofimova et al., 2018*; *Wang et al., 2017*). However, these studies primarily focus on stabilized microtubules, short protofilaments, or tubulin dimers, leaving the mechanism by which MCAK binds to growing microtubule ends remaining elusive.

In this study, we use single-molecule imaging to investigate the direct binding of MCAK to growing microtubule ends. A key finding is that MCAK binds to the proximal region of GTP cap where GDP·Pi-tubulins accumulate, in addition to its known binding at the distalmost tip, where XMAP215 binds to curved protofilaments. Further analysis reveals that MCAK strongly binds to GTPγS microtubules, which are thought to mimic GDP·Pi-tubulins enriched in the EB cap. These findings indicate that MCAK recognizes the nucleotide-dependent feature of growing microtubule ends. Having confirmed that MCAK binds to the entire GTP cap structure, we next examine how MCAK co-regulates microtubule dynamics in collaboration with XMAP215 and reveal their separate effect on growth rate and catastrophe frequency, respectively. Overall, this study provides novel insights into how MCAK binds to and acts at growing microtubule ends.

## Results

### MCAK shows a binding preference for growing microtubule ends

We began by measuring the single-molecule binding kinetics of MCAK at growing microtubule ends using the in vitro microtubule dynamics assay. This experiment was performed in BRB80 supplemented with 50 mM KCl and 1 mM ATP, providing a near-physiological ion strength of approximately 0.2 M, similar to what was previously reported (*Scopes, 2002*; *Storey, 2004*). The size-exclusion chromatography analysis showed that the purified GFP-MCAK exists as dimers under this condition (*Figure 1—figure supplement 1*).

We observed MCAK binding events both at growing microtubule ends and along the lattice (*Figure 1A*). To characterize these events, we measured the fluorescence intensity of each binding event (see Materials and methods) and plotted a probability distribution of all binding events (*Figure 1—figure supplement 1*). We used the range of fluorescence intensity from the probability distribution (μ±2σ: 300±145 A.U.) to identify single MCAK dimers. Using a custom software (*Maurer et al., 2014*; *Song et al., 2020*), we determined the end-binding position of each individual MCAK molecule in each frame by fitting a Gaussian function to the intensity profile along the longitudinal axis of the microtubule, with the plus-end serving as the positional reference (*Figure 1B*). The average position over the dwell time of each MCAK molecule was taken as the position of this binding event

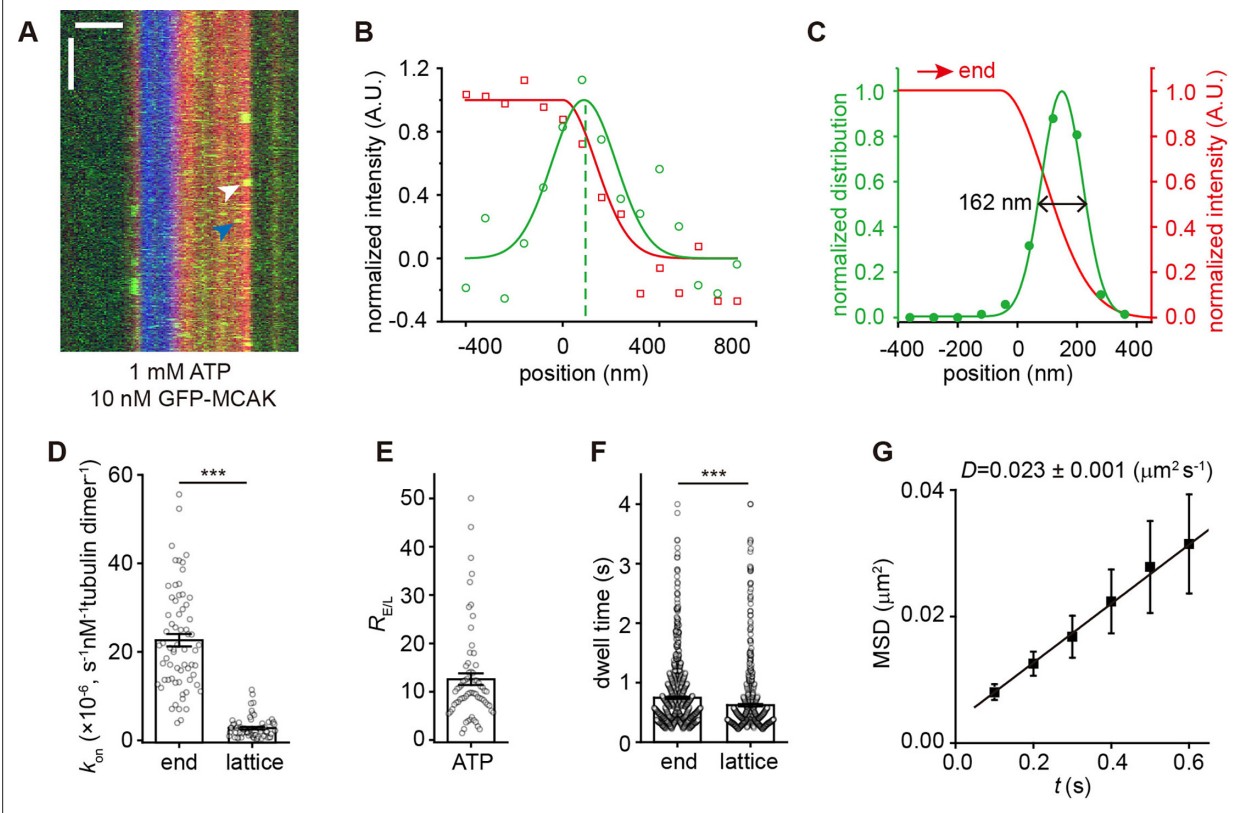

**Figure 1.** MCAK preferentially binds to growing microtubule ends. (**A**) Representative kymographs of single-molecule GFP-MCAK (green, 10 nM, measured in terms of monomer concentration unless otherwise stated) binding events on dynamic microtubules (red, tubulin: 16 mM) growing from GMPCPP microtubule seeds (blue) in the presence of 1 mM ATP. A plus-end binding event and a lattice binding event were marked by the white and blue arrowheads, respectively. Scale bars: vertical, 5 s; horizontal, 2 µm. (**B**) Representative intensity profiles of a single GFP-MCAK molecule (green, 10 nM) and a growing microtubule end (red, tubulin: 16 mM) at one frame during the dwell time of a binding event (corresponding to the white arrowhead in panel **A**). The peak position of a GFP-MCAK molecule was determined by Gaussian fitting (green dashed line). (**C**) Position probability distributions of GFP-MCAK binding sites (green, n=152 events) along the microtubule longitudinal axis. The red curve represents the fitting curve of the growing microtubule end as the positional reference. The MCAK binding region was defined as the area within the FWHM (double black arrow, 162 nm) of the fitted green curve. (**D**) Statistical quantification of on-rate ($k_{on}$) of GFP-MCAK's binding to the plus-end versus lattice of dynamic microtubules in the presence of 1 mM ATP (n=66 microtubules from 3 assays). (**E**) Statistical quantification of $R_{E/L}$ in the presence of 1 mM ATP (n=66 microtubules from 3 assays). (**F**) Statistical quantification of the dwell time of GFP-MCAK's binding to the growing end versus lattice of dynamic microtubules in the presence of 1 mM ATP (n=966 events from 3 assays for the plus end; n=702 events from 3 assays for lattice). (**G**) Mean-squared displacement (MSD) of GFP-MCAK plotted against the time interval ($t$, 0.1 s per frame). The diffusion coefficient (**D**) was calculated via linear regression of the equation ($x^2 \geq 2 Dt$), yielding 0. 023 µm²s⁻¹ (error bars represent SEM, n=203 trajectories). In panels **D**, **E**, and **F**, all the data were presented as mean ± SEM. All comparisons were performed using two-tailed Mann-Whitney $U$ test with Bonferroni correction, n.s., no significance; *, p<0.05; **, p<0.01; ***, p<0.001.

The online version of this article includes the following source data and figure supplement(s) for figure 1:

**Source data 1.** Numerical data used to generate *Figure 1*.

**Figure supplement 1.** Biochemical and functional characterization of purified MCAK, its variants, EB1, and XMAP215.

**Figure supplement 1—source data 1.** Numerical data used to generate *Figure 1—figure supplement 1*.

**Figure supplement 1—source data 2.** Original files for the western blot and the gel displayed in *Figure 1—figure supplement 1B and J*.

**Figure supplement 1—source data 3.** PDF file containing original western blot and the gel for *Figure 1—figure supplement 1B and J*, indicating the relevant bands.

**Figure supplement 2.** Single-molecule localization analysis at growing microtubule ends.

**Figure supplement 2—source data 1.** Numerical data used to generate *Figure 1—figure supplement 2*.

(*Figure 1—figure supplement 2*). From this, we generated a probability distribution of all binding positions, derived from 152 binding events across three assays (*Figure 1C*). This analysis revealed that the MCAK binding region was 162 nm in length (FWHM, Full Width at Half Maximum), corresponding to approximately 20 layers of tubulin dimers. We categorized the binding events of MCAK within this region as 'end binding' and those located more proximally as 'lattice binding'.

To investigate the binding kinetics, we first compared the apparent association rates of MCAK on the plus-end ($k_{on-P}$) and the lattice ($k_{on-L}$) in the presence of ATP. The ratio ($R_{E/L}$) of $k_{on-P}$ ((22. 6±1.4)×10$^{-6}$ s$^{-1}$ nM$^{-1}$, per tubulin dimer, mean ± SEM, hereafter unless otherwise specified) to $k_{on-L}$ ((2.8±0.3)×10$^{-6}$ s$^{-1}$ nM$^{-1}$) was 13±1 (n=66 microtubules from 3 assays; *Figure 1D and E*). We also measured the dwell time, which reflects the dissociation rate ($k_{off}$) (*Figure 1—figure supplement 1*). MCAK exhibited a longer dwell time at the plus-end than on the lattice (plus end: 0.75±0.02 s, n=966 events from 3 assays; lattice: 0.62±0.02 s, n=702 events from 3 assays, p<0.001; *Figure 1F*). Using these kinetic parameters, we calculated the dissociation constant ($K_d$) of MCAK for individual tubulin dimers at the plus-end as 69 µM. This affinity may appear relatively low, but when considering the microtubule end as a multivalent receptor with multiple MCAK-binding sites (end length = 160 nm) - where only a few bound MCAK molecules are required to regulate microtubule dynamics parameters - the effective dissociation constant reaches the nanomolar range (260 nM). The MCAK's lattice binding affinity was even lower ($K_{d-L}$=1057 µM, per tubulin dimer), demonstrating that MCAK has a clear end-binding preference. We also observed single-molecule binding events of MCAK at the minus-ends (*Figure 1A*), with a similar $R_{E/L}$ (13±2, n=28 microtubules from 3 assays) and $K_d$ (40 µM, per tubulin dimer). We focused on the plus-end binding events (referred to as 'end-binding' hereafter), due to their greater physiological relevance.

Having measured the single-molecular binding events, we next investigated whether the binding of MCAK to growing microtubule ends occurs through direct end-binding (3D diffusion) or 1D lattice-diffusion in our experimental conditions (*Desai et al., 1999*; *Montenegro Gouveia et al., 2010*; *Helenius et al., 2006*; *Oguchi et al., 2011*). Note that these two mechanisms are not mutually exclusive. Our data revealed that the lattice-diffusion coefficient of MCAK ($D$) on the lattice of dynamic microtubules was 0.023±0.001 µm$^2$·s$^{-1}$ (*Figure 1G*), comparable to the values reported under high-salt conditions (*Cooper et al., 2010*; *Montenegro Gouveia et al., 2010*), but lower than that in low-salt conditions (*Helenius et al., 2006*). Because the dwell time of the lattice-binding events was approximately 0.6 s, the average length scanned by a MCAK molecule via 1D lattice-diffusion was ~160 nm, roughly the length of GTP cap (*Guesdon et al., 2016*; *Maurer et al., 2014*). Using an established method (*Helenius et al., 2006*), we calculated the flux of MCAK arriving at growing ends via 1D diffusion was 0.004 s$^{-1}$. This was significantly lower than the observed MCAK arrival rate at the plus-end observed ($N_p$ = $k_{on}$·$C$=0.067 s$^{-1}$. $C$: MCAK concentration). Therefore, in our experimental conditions, 1D lattice-diffusion contributes minimally to the growing end-binding of MCAK (~6%), suggesting that the end-binding primarily results from the direct binding pathway.

## MCAK binds to the EB cap

We sought to understand how MCAK recognizes growing microtubule ends. Previous studies, mostly based on tubulin dimers, short protofilaments, or stabilized microtubules, suggested that MCAK binds microtubule ends by recognizing curved protofilaments (*Asenjo et al., 2013*; *Benoit et al., 2018*; *Moores and Milligan, 2008*; *Tan et al., 2008*; *Trofimova et al., 2018*). This model predicts that MCAK binds to the distalmost end of growing microtubules, overlapping with the binding region of XMAP215 family members (i.e. the distalmost cap; *Ayaz et al., 2012*; *Brouhard et al., 2008*). To test this hypothesis, we quantified the end-binding regions of MCAK, EB1, and XMAP215 (*Figure 2A and B*) by measuring the FWHM of their spatial distribution. To our surprise, the probability distribution of MCAK's positions exhibited a significant difference from that of XMAP215 (p<10$^{-5}$, Kolmogorov-Smirnov test). The end-binding region of MCAK (FWHM = 185 nm) was longer than that of XMAP215 (FWHM = 123 nm; *Figure 2B*). It extended proximally towards the lattice, covering the binding region of EB1 (i.e. the EB cap, FWHM = 118 nm). To further understand whether MCAK binds to the distalmost cap and the EB cap with different kinetics, we compared the dwell time of MCAK at the distal and proximal locations (*Figure 2B*). As shown in *Figure 2B*, the MCAK molecules located to the left of the XMAP215 binding region were classified as proximally distributed, while those located to the right of the EB1 binding region were considered distally distributed. No significant difference was

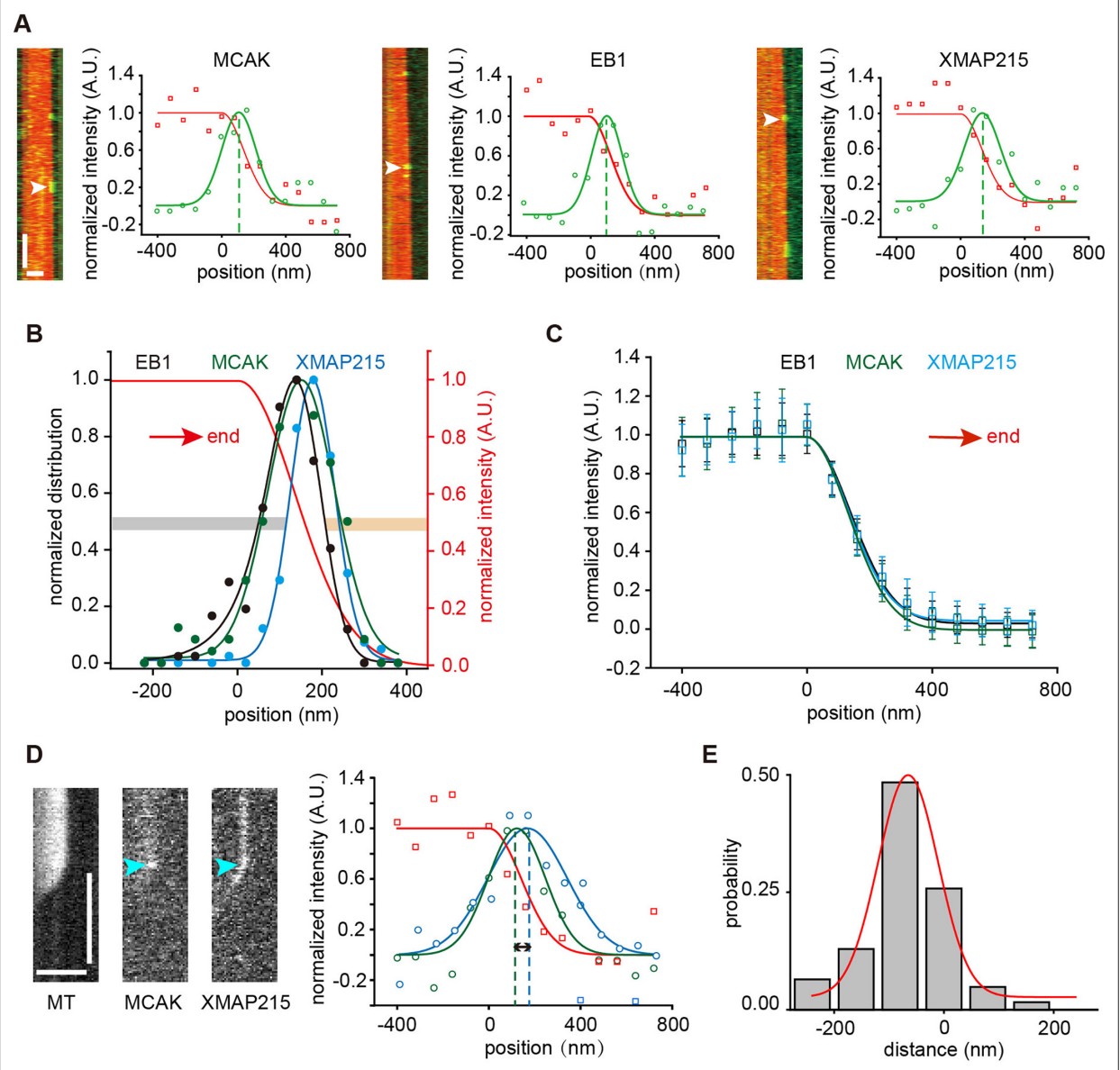

**Figure 2.** MCAK binds to the EB cap of growing microtubule ends. (**A**) Representative kymographs showing the individual binding events of GFP-MCAK (1 nM, with 1 mM ATP, left), EB1-GFP (10 nM, middle), or XMAP215-GFP (1 nM, right) at growing microtubule ends (red, tubulin: 12 µM). The plots showed the representative intensity profiles of single molecules (green) and growing microtubule ends (red) at one frame during the dwell time of the individual binding event (white arrowheads). Peak positions were determined by Gaussian fitting (green dashed line). Scale bars: vertical, 5 s; horizontal, 2 µm. (**B**) Position probability distributions of the binding sites of GFP-MCAK (green, n=123 events), EB1-GFP (black, n=184 events) and XMAP215-GFP (blue, 142 events) along the microtubule longitudinal axis. The fitting curve of the growing microtubule end (red) was used as the positional reference. The binding region was defined as the region within the FWHM of the fitted curve. The GFP-MCAK molecules localized to the left of XMAP215-GFP's binding region were defined as proximally distributed (gray bar), and those localized to the right of EB1-GFP's binding region were defined as distally distributed (orange bar). (**C**) Averaged intensity profiles of the growing microtubule ends decorated with GFP-MCAK (green, 123 microtubules), EB1-GFP (black, 184 microtubules) and XMAP215-GFP (blue, 142 microtubules). Data were presented as mean ± std, and no significant difference was observed between the three conditions (Kolmogorov-Smirnov test, $End_{EB1}$ vs $End_{MCAK}$, p=0.68; $End_{MCAK}$ vs $End_{XMAP215}$, p=0.68; $End_{EB1}$ vs $End_{XMAP215}$, p=1). In panels **B** and **C**, the red arrows indicated microtubule growth direction. (**D**) Representative kymographs showing concurrent binding events of XMAP215-GFP (25 nM) and MCAK-RFP (20 nM) at growing microtubule ends (tubulin: 12 µM) in the presence of 1 mM ATP. The plots showed the representative intensity profiles of XMAP215-GFP (blue), MCAK-RFP (green), and growing microtubule ends (red) at one frame during the dwell time of the binding event (cyan arrowhead). Peak positions were determined by Gaussian fitting (dashed line). The spacing between the two peaks represents the instantaneous distance between MCAK and XMAP215 (double black arrow). Scale bars: vertical, 5 s; horizontal, 2 µm. (**E**) Probability distribution of the distance between XMAP215-GFP and MCAK-RFP (62 events from 3 assays) at growing microtubule ends. In this plot, the position of XMAP215-GFP was taken as the origin. A negative distance indicated a more proximal localization.

*Figure 2 continued on next page*

*Figure 2 continued*

The online version of this article includes the following source data for figure 2:

**Source data 1.** Numerical data used to generate *Figure 2*.

found between the two groups ($\tau_{distal}$=0.8 ± 0.1 s, n=28 binding events; $\tau_{proximal}$=0.7±0.1 s, n=45 binding events; p>1, the two-tailed Mann-Whitney U test with Bonferroni correction). Additionally, no significant differences were observed in the intensity profiles of the microtubule ends in the presence of MCAK (1 nM), EB1 (10 nM), and XMAP215 (1 nM), respectively ($End_{EB1}$ vs $End_{MCAK}$, p=0.68; $End_{MCAK}$ vs $End_{XMAP215}$, p=0.68; $End_{EB1}$ vs $End_{XMAP215}$, p=1. Kolmogorov-Smirnov test) (*Figure 2C*).

The results suggest that MCAK's end-binding region partially overlaps with that of XMAP215, with more MCAK positioned behind XMAP215, closer to the lattice. To confirm this, we assessed the colocalization of MCAK and XMAP215. Given the low probability of both molecules simultaneously appearing at microtubule ends at low concentrations, we increased the concentration of XMAP215 to ensure its consistent tracking of growing microtubule ends while simultaneously recording single-molecule binding events of MCAK. In each frame, we measured the distance between the intensity peaks of XMAP215 and MCAK along the microtubule's longitudinal axis (*Figure 2D*). The average distance over MCAK's binding duration was then calculated to determine its relative position to XMAP215. Subsequently, we plotted a probability distribution of all relative distances (*Figure 2E*). This distribution peaked at –65 nm, indicating that MCAK predominantly binds behind XMAP215, with 14.5% of binding events occurring within XMAP215's binding region (distance >0). These findings support the notion that MCAK not only binds to the distalmost tip but also to the relatively proximal region, the EB cap.

## The end-binding preference of MCAK is nucleotide state-dependent

It has been reported that MCAK tightly binds to microtubules in both the AMPPNP (a non-hydrolysable ATP analogue, mimicking the ATP-bound state) and nucleotide-free states, but exhibits weaker binding in the ADP-bound state (*Friel and Howard, 2011*; *Helenius et al., 2006*; *Wang et al., 2012*). Moreover, kinesin-13s in the AMPPNP state preferentially form protofilament curls or rings with tubulins (*Asenjo et al., 2013*; *Benoit et al., 2018*; *Tan et al., 2008*). These findings suggest that the interaction between kinesin-13s and microtubules is regulated by their nucleotide-bound state. Given this, we sought to determine whether the end-binding preference of MCAK is also nucleotide state-dependent. To explore this, we measured the end-binding kinetics of MCAK in complex with AMPPNP, ADP, or in the nucleotide-free (APO) state (*Figure 3A*), under conditions that maintain similar microtubule growth rates (*Figure 1—figure supplement 1*).

Comparing the binding kinetics of MCAK in different nucleotide states, we found that $k_{on-P}$ and $k_{on-L}$ of MCAK·AMPPNP were both lower than those of MCAK·ATP (*Figure 3B and C*), yet resulting in a similar end-binding preference ($R_{E/L}$=11 ± 2, n=29 microtubules from 2 assays, p>1). However, the binding of MCAK·AMPPNP at the end and lattice both showed a longer dwell time compared to that of MCAK·ATP (3.32±0.24 s at the end, n=231 events from 2 assays, p<0.001; 1.46±0.21 s at lattice, n=142 events from 2 assays, p<0.001), indicating lower $k_{off}$ (*Figure 3D* and *Figure 1—figure supplement 1*). As a result of the proportionally reduced $k_{on-P}$ and $k_{off}$, the end-binding affinity of MCAK·AMPPNP ($K_{d-P}$=61 µM, per tubulin dimer, hereafter) was comparable to that of MCAK·ATP. In contrast, MCAK·ADP showed nearly unchanged $k_{on-P}$ but increased $k_{on-L}$, leading to a lower $R_{E/L}$ (2±0.3, n=21 microtubules from 2 assays, p<0.001) and suggesting the reduction, but not a complete loss, of the end-binding preference (*Figure 3B and C*). Additionally, MCAK·ADP showed a significantly shorter end-binding dwell time (0.46±0.02 s at the end, n=327 events from 2 assays, p<0.001) compared to MCAK·ATP (*Figure 3D*). Consequently, MCAK·ADP has a lower end-binding affinity ($K_{d-P}$=163 µM). MCAK·APO, the nucleotide-free state, showed a large increase in $k_{on-P}$ and $k_{on-L}$, thereby having a lower $R_{E/L}$ (2±0.2, n=36 microtubules from 3 assays, p<0.001; *Figure 3B and C*). It also showed a significantly longer end-binding dwell time (1.89±0.23 s, n=71 events from 3 assays, p<0.001; *Figure 3D*), indicating a lower $k_{off}$ (*Figure 1—figure supplement 1*). As a result, MCAK·APO had a higher end-binding affinity ($K_{d-P}$=1.8 µM), but it partially loses the end-binding preference. Overall, these findings indicate that MCAK's end-binding preference and affinity are dependent on its nucleotide state, with the ATP-bound state exhibiting the highest end-binding preference.

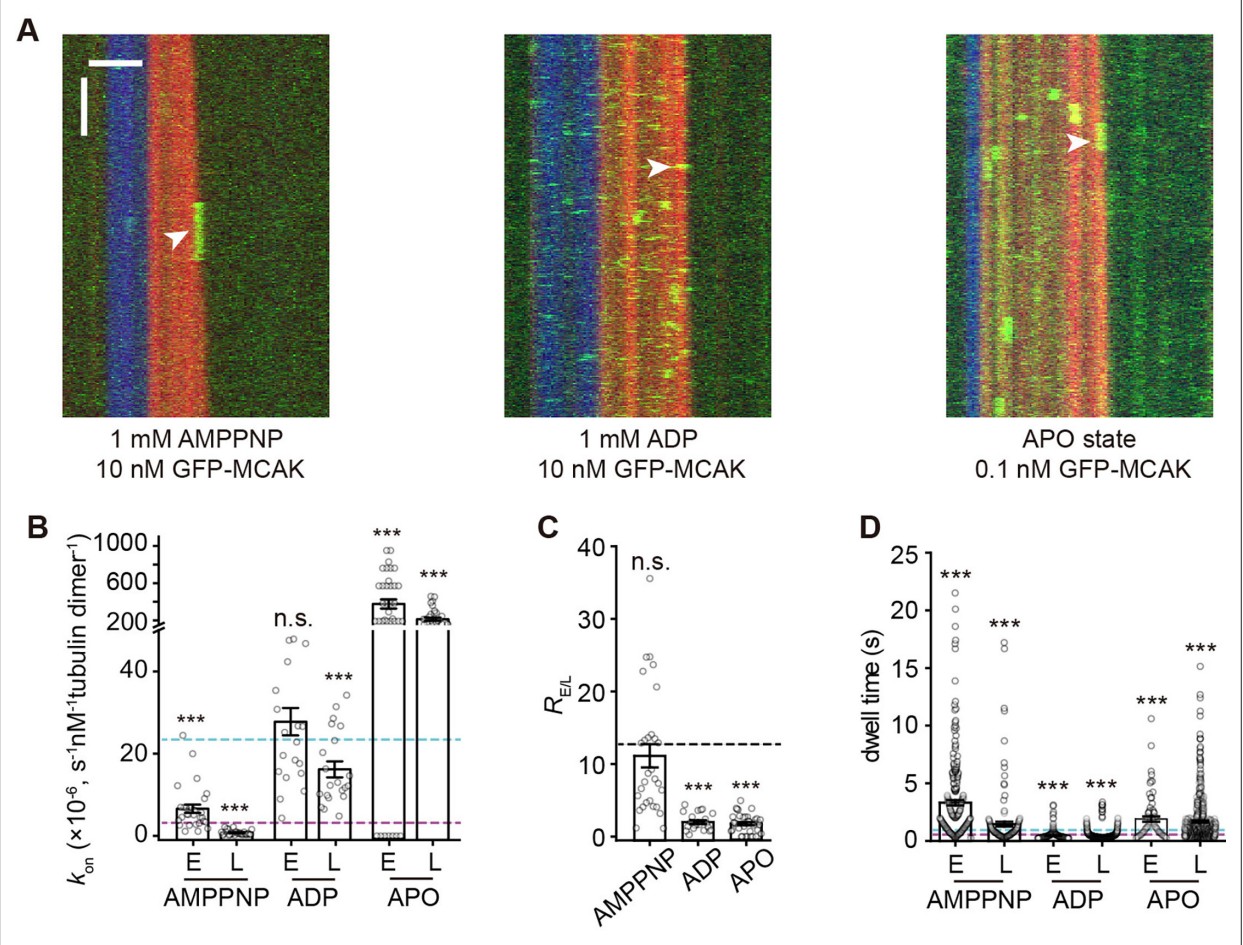

**Figure 3.** The growing end-binding preference of MCAK is dependent on its nucleotide state. (**A**) Representative kymographs of single-molecule GFP-MCAK (green) binding events on dynamic microtubules (red, tubulin: 16 µM) growing from the GMPCPP microtubule seeds (blue) in the presence of 1 mM AMPPNP (left), 1 mM ADP (middle), and APO (right; the nucleotide-free state). The plus-end binding events were indicated by white arrowheads. Scale bar: vertical, 5 s; horizontal, 2 µm. (**B**) Statistical quantification of on-rate ($k_{on}$) of GFP-MCAK binding to the plus end (**E**) versus the lattice (**L**) of dynamic microtubules in the presence of 1 mM AMPPNP (n=29 microtubules from 2 assays), 1 mM ADP (n=21 microtubules from 2 assays), and APO (n=36 microtubules from 3 assays). The cyan and purple dashed lines denoted the on-rate value of MCAK binding at the plus end and lattice in the ATP condition (from *Figure 1*), respectively. (**C**) Statistical quantification of $R_{E/L}$ in the presence of 1 mM AMPPNP (n=29 microtubules from 2 assays), 1 mM ADP (n=21 microtubules from 2 assays), and APO (n=36 microtubules from 3 assays). Dashed line: the mean value of $R_{E/L}$ in the ATP condition from *Figure 1*. (**D**) Statistical quantification of the dwell time for GFP-MCAK binding at the growing ends and lattice of dynamic microtubules in the presence of 1 mM AMPPNP (n=231 events from 2 assays for the plus end; n=142 events from 2 assays for lattice), 1 mM ADP (n=327 events from 2 assays for the plus end; n=1184 events from 2 assays for lattice), and APO (n=71 events from 3 assays for the plus end; n=573 events from 3 assays for lattice). The cyan and purple dashed lines denoted the dwell time value of MCAK binding at the plus end and lattice in the ATP condition (from *Figure 1*), respectively. In the panels **B**, **C**, and **D**, all the data were presented as mean ± SEM. All the comparisons were made against the corresponding value in the ATP condition from *Figure 1* using two-tailed Mann-Whitney U test with Bonferroni correction, n.s., no significance; *, p<0.05; **, p<0.01; ***, p<0.001.

The online version of this article includes the following source data for figure 3:

**Source data 1.** Numerical data used to generate *Figure 3*.

## MCAK strongly binds to GTPγS microtubules

The binding region of MCAK at growing microtubule ends encompasses the EB cap. This region consists mostly of GDP·Pi-tubulin dimers, which are thought to form at least some lateral contacts and adopt a straighter conformation (*Guesdon et al., 2016*; *Maurer et al., 2012*). This raises the question of how MCAK recognizes the EB cap.

We first investigated the contribution of the nucleotide state-dependent feature of the EB cap, which underlies the end-binding preference of EBs (*Maurer et al., 2011*; *Maurer et al., 2012*). Specifically, we compared the binding affinities of MCAK·ATP on GTPγS and GDP microtubules, mimicking

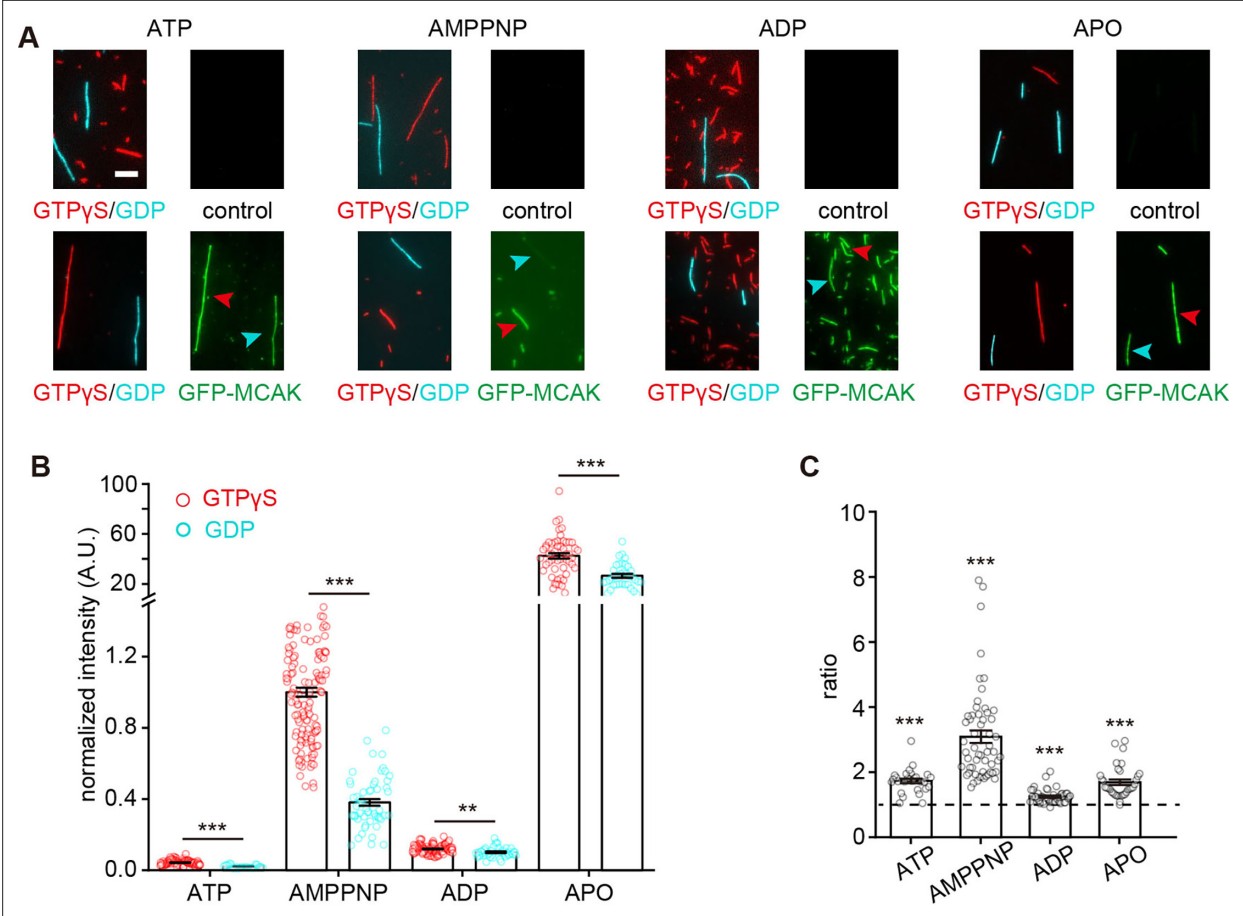

**Figure 4.** MCAK strongly binds to GTPγS microtubules in a nucleotide-dependent manner. (**A**) Representative projection images of GFP-MCAK binding to GTPγS microtubules (red) versus GDP microtubules (cyan) in the presence of 1 mM ATP (left, 10 nM GFP-MCAK), 1 mM AMPPNP (left middle, 1 nM GFP-MCAK), 1 mM ADP (right middle, 10 nM GFP-MCAK) and at the APO state (right, 0.1 nM GFP-MCAK). Note that the binding on the GDP microtubules was compared to that on the GTPγS microtubules in the same flow cell. The GFP-MCAK binding to the GTPγS and GDP microtubules was indicated by the red and cyan arrowhead, respectively. Scale bar: 5 µm. (**B**) Statistical quantification of the normalized fluorescence intensity of GFP-MCAK (1 nM) on different microtubules in the presence of 1 mM ATP (95 GTPγS microtubules, 28 GDP microtubules from 2 assays), 1 mM AMPPNP (133 GTPγS microtubules, 56 GDP microtubules from 3 assays), 1 mM ADP (111 GTPγS microtubules, 44 GDP microtubules from 2 assays), or at the APO state (52 GTPγS microtubules, 36 GDP microtubules from 3 assays). All data were normalized to the binding to GTPγS microtubules binding in the AMPPNP condition. (**C**) The binding intensity ratios of GFP-MCAK at GTPγS microtubules to GDP microtubules under various nucleotide conditions. ATP: n=28 GTPγS/GDP from 2 assays. AMPPNP: n=56 GTPγS/GDP from 3 assays. ADP: n=44 GTPγS/GDP from 2 assays. APO: n=33 GTPγS/GDP from 3 assays in APO state. The dashed line represented 1. Statistical comparisons were performed between the ratios and 1. In panels **B** and **C**, all the data were presented as mean ± SEM. All comparisons were performed using two-tailed Mann–Whitney U test with Bonferroni correction, n.s., no significance; *, p<0.05; **, p<0.01; ***, p<0.001.

The online version of this article includes the following source data and figure supplement(s) for figure 4:

**Source data 1.** Numerical data used to generate *Figure 4*.

**Figure supplement 1.** Projection of single-molecule fluorescence images.

**Figure supplement 2.** EB1-GFP strongly binds to GTPγS microtubules.

**Figure supplement 2—source data 1.** Numerical data used to generate *Figure 4—figure supplement 2*.

the tubulin dimers in the EB cap and the lattice of growing microtubules, respectively (*Figure 4A*). Because MCAK·ATP at the concentration required for full lattice decoration would rapidly disassemble microtubules, we cannot directly compare the total binding of MCAK on microtubules as what was previously done for the EBs (*Maurer et al., 2011*; *Maurer et al., 2012*). Instead, we adopted an alternative approach by recording the binding of MCAK·ATP at a low concentration (10 nM) on microtubules and measuring fluorescence intensity summation over a period of time (1000 frames, 0.3 s per frame; *Figure 4—figure supplement 1*). Strikingly, quantitative analysis revealed that MCAK·ATP

showed a significantly stronger binding to GTPγS microtubules than to GDP microtubules (*Figure 4B and C*). Because the tubulin dimers in GTPγS microtubules were considered to be the analogue of GDP·Pi-tubulins in the EB cap (*Manka and Moores, 2018*; *Maurer et al., 2011*; *Maurer et al., 2014*; *Zhang et al., 2015*), this result provides a biochemical basis for the MCAK·ATP's binding preference for the EB cap. Here, we used EB1 as a control (*Figure 4—figure supplement 2*).

Given that the growing end-binding affinity and preference of MCAK depend on the nucleotide state, we wondered whether its preference for GTPγS microtubules follows a similar trend. To address this, we performed the experiments for MCAK·AMPPNP, MCAK·ADP, and MCAK·APO. The MCAK variants bound to the lattice of microtubules with different affinities (MCAK·APO >MCAK·AMPPNP >MCAK·ADP >MCAK·ATP; *Figure 4B*), consistent with the results observed on the lattice of dynamic microtubules. Additionally, MCAK maintained a binding preference for GTPγS microtubules over GDP microtubules across all nucleotide states, but the degree of the preference varied with the nucleotide state (MCAK·AMPPNP >MCAK·ATP ≈ MCAK·APO >MCAK·ADP; *Figure 4C*). This finding indicates that MCAK's binding preference for GTPγS microtubules (i.e. GDP·Pi-tubulin) is strongest in the ATP state, mirroring its preference for growing microtubule ends.

## MCAK binds to the end region of stabilized GMPCPP microtubules in a nucleotide state-dependent manner

Our findings also support the notion that MCAK binds to the distalmost cap, consistent with the previous reports that MCAK binds to the curved protofilaments of microtubule ends (*Asenjo et al., 2013*; *Benoit et al., 2018*; *Desai et al., 1999*; *Tan et al., 2008*; *Wang et al., 2017*). We then questioned whether the binding preference for curved protofilaments also depends on the nucleotide state of MCAK. To investigate this issue, we analyzed the binding of MCAK at the ends of stabilized GMPCPP microtubules (*Figure 5A*), where curved protofilaments are thought to resemble those at the ends of dynamic microtubules, but without the presence of GDP·Pi-tubulins (*Atherton et al., 2018*; *Manka and Moores, 2018*; *McIntosh et al., 2018*). Taxol-stabilized GDP-microtubules were here used as a control (*Figure 5—figure supplement 1*). We found that MCAK, in both the ATP and AMPPNP state, exhibited a binding preference for the ends of GMPCPP-stabilized microtubules (*Figure 5A–F*). Similarly, MCAK·ATP showed a binding preference for the ends of taxol-stabilized GDP microtubules, albeit to a lesser extent (*Figure 5—figure supplement 1*). This observation suggests that the ends of stable microtubules mimic the structure of growing microtubule ends to varying degrees, with the GMPCPP-stabilized ends providing the closest resemblance. In contrast, MCAK·APO and MCAK·ADP did not show the end-binding preference (*Figure 5A–F*), consistent with previous reports that MCAK in the two nucleotide state binds weakly to the curved protofilament or tubulin dimer (*Asenjo et al., 2013*; *Wang et al., 2012*). Additionally, it also corroborates our observation that MCAK·ADP and MCAK·APO had a lower binding preference for growing microtubule ends.

## The α4 helix and the L2 loop of MCAK contribute to the binding preference to the EB cap

Intuitively, the binding preference for the EB cap, in addition to the previously established preference for the curved protofilaments, provides more binding sites for MCAK at growing microtubule ends. We sought to identify the key domains responsible for MCAK's preference for the EB cap. Previous studies have shown that the α4 helix and the L2 loop are critical for mediating MCAK's interaction with curved protofilaments (*Asenjo et al., 2013*; *Benoit et al., 2018*; *Patel et al., 2016*; *Tan et al., 2008*; *Wang et al., 2017*). Specifically, the α4 helix of MCAK interacts with the intradimer interface of the tubulin heterodimer, while the L2 loop contacts the interdimer interface (*Figure 6A*). Moreover, the electrostatic interactions between the α4 helix and the L2 loop of MCAK with the tubulin heterodimer are important for MCAK's interactions with microtubule ends (*Patel et al., 2016*; *Tan et al., 2008*), as phosphorylation at T537 disrupts MCAK's microtubule-end binding and depolymerization activity (*Sanhaji et al., 2010*). We then wondered whether these domains also contribute to the binding preference for the EB cap. To test this hypothesis, we introduced two mutations: the K524A mutation in the α4 helix, which is expected to disrupt the electrostatic interaction between MCAK and the tubulin heterodimer (*Patel et al., 2016*; *Wang et al., 2017*), and the V298S mutation in the L2 loop, which is predicted to impair the hydrophobic interaction between the L2 loop and α-tubulin (*Wang et al., 2015*). Consistent with previous reports (*Wang et al., 2017*; *Wang et al., 2015*), both mutants

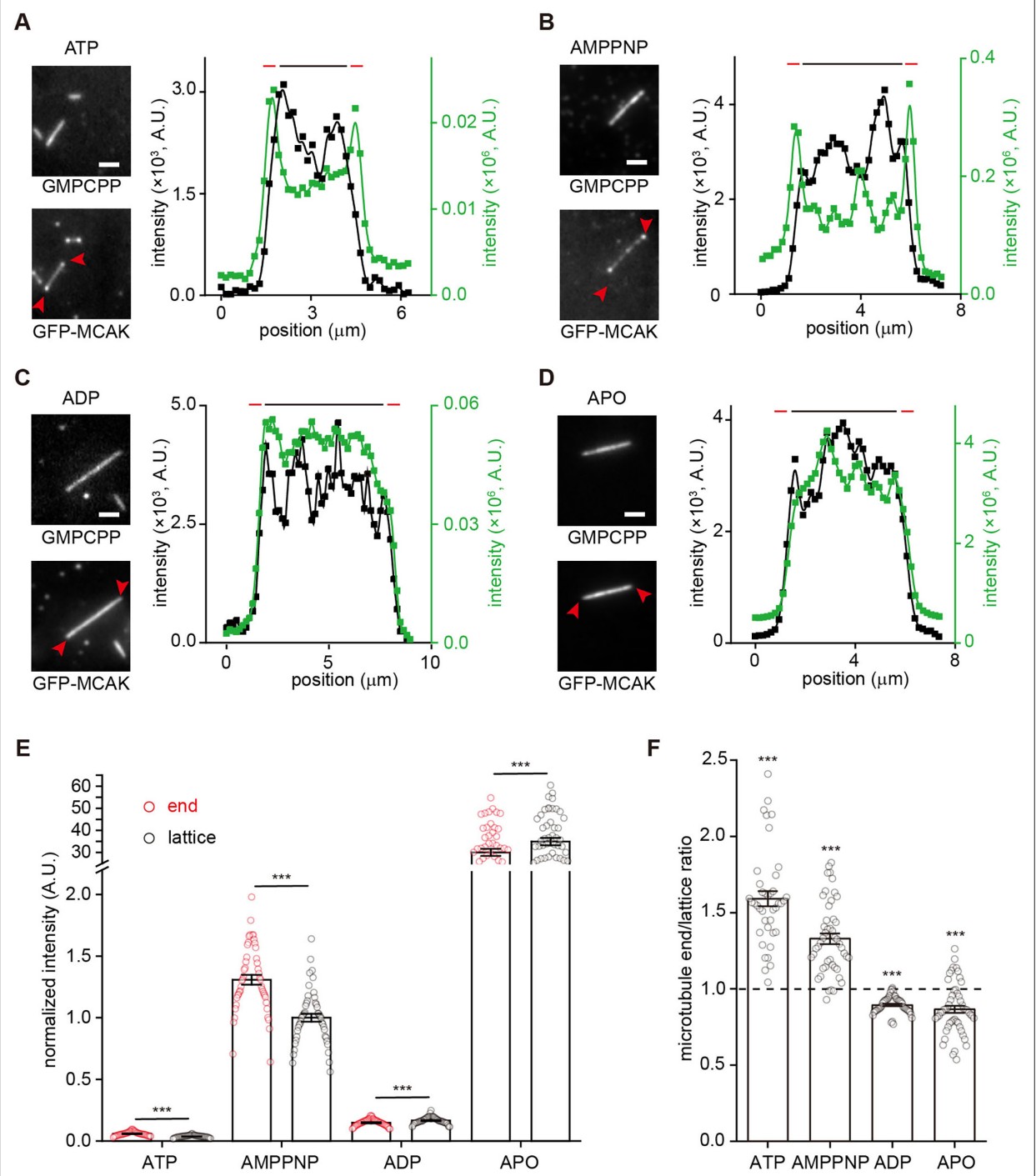

**Figure 5.** MCAK binds to the ends of GMPCPP microtubules in a nucleotide state-dependent manner. (**A–D**) Representative fluorescence projection images and intensity profiles showing GFP-MCAK binding to the ends and lattice of GMPCPP microtubules in the presence of 1 mM ATP (10 nM GFP-MCAK), 1 mM AMPPNP (1 nM GFP-MCAK), 1 mM ADP (10 nM GFP-MCAK), or at the APO state (0.1 nM GFP-MCAK). Red arrowhead: the signal of MCAK enriching at the ends. Intensity profiles (right panels) showed spatial intensity signal distributions of the microtubule (black curve) and GFP-MCAK (green curve). Red and black bars indicated the end and the lattice, respectively. Scale bar = 2 µm. (**E**) Statistical quantification of the normalized binding intensity of GFP-MCAK (1 nM) at the end versus lattice regions of GMPCPP microtubules in the presence of 1 mM ATP (40 microtubules from 3 assays), 1 mM AMPPNP (47 microtubules from 3 assays), 1 mM ADP (45 microtubules from 3 assays), or at the APO state (52 microtubules from 3 assays). All the data were normalized to the binding intensity of GFP-MCAK at GMPCPP microtubule lattices in the AMPPNP condition. All the data were presented as mean ± SEM. Statistical analysis was performed using two-tailed paired *t*-test by Bonferroni correction, ***, p<0.001. (**F**) The end-to-lattice binding intensity ratios of GFP-MCAK at GMPCPP microtubules under various nucleotide conditions, 1 mM ATP (40 microtubules from 3 assays), 1 mM AMPPNP

*Figure 5 continued on next page*

*Figure 5 continued*

(47 microtubules from 3 assays), 1 mM ADP (45 microtubules from 3 assays), or the APO state (52 microtubules from 3 assays). The ratios were presented as mean ± SEM. The dashed line represents 1. Statistical comparisons were performed between the ratios and 1 using two-tailed Mann-Whitney U test with Bonferroni correction, n.s., no significance; *, p<0.05; **, p<0.01; ***, p<0.001.

The online version of this article includes the following source data and figure supplement(s) for figure 5:

**Source data 1.** Numerical data used to generate *Figure 5*.

**Figure supplement 1.** MCAK preferentially binds to the ends of taxol-stabilized GDP microtubules.

**Figure supplement 1—source data 1.** Numerical data used to generate *Figure 5—figure supplement 1*.

exhibited deficiencies in depolymerization and reduced ATPase activity (*Figure 6—figure supplement 1*).

Compared to the wild-type MCAK, both MCAK$^{K524A}$ and MCAK$^{V298S}$ were capable of binding to growing microtubule ends (*Figure 6B*), but their $k_{on-P}$ and dwell time at growing microtubule ends were significantly reduced (*Figure 6C and D*). This indicates that the binding affinity of the mutants to growing microtubule ends is substantially decreased. Here, due to the minimal to negligible number of observed binding events of the mutants at growing microtubule ends and the lattice, reliable calculations of $R_{E/L}$ and $k_{off}$ were not feasible for the mutants. Therefore, we directly assayed the binding preference for GTPγS microtubules mimicking the EB cap. Both mutants showed significantly reduced binding intensity and a decreased, but not the loss of, preference for GTPγS microtubules compared to wild-type MCAK (*Figure 6E–G* vs *Figure 4B and C*). These results suggest that both the α4 helix and the L2 loop contribute to the binding preference for the EB cap.

## Functional specification of MCAK and XMAP215

MCAK is a microtubule catastrophe factor, whereas XMAP215 functions as a microtubule polymerase (*Brouhard et al., 2008*; *Desai et al., 1999*; *Gardner et al., 2011*). Both are capable of binding to dynamic microtubule ends and regulating microtubule dynamics. In the previous reports, they were shown to antagonistically regulate microtubule assembly (*Barr and Gergely, 2008*; *Kinoshita et al., 2001*; *Moriwaki and Goshima, 2016*; *Tournebize et al., 2000*). Additionally, structural models have indicated that both MCAK and XMAP215 bind to curved tubulin dimers (*Ayaz et al., 2012*; *Trofimova et al., 2018*), raising the possibility of competition for binding sites in the distalmost cap. However, while XMAP215 was reported to promote microtubule catastrophe (*Farmer et al., 2021*; *Holmfeldt et al., 2004*; *Vasquez et al., 1994*), our findings showed that MCAK binds more proximally relative to the binding region of XMAP215 at growing microtubule ends. The extent to which competition from XMAP215 would antagonize the function of MCAK remains to be elucidated. Therefore, the collective effect of MCAK and XMAP215 on microtubule dynamics remains an intriguing question.

To investigate this issue, we studied how MCAK and XMAP215 together regulate microtubule growth rate and catastrophe frequency. Based on our pilot experiments, we chose 20 nM for MCAK and 50 nM for XMAP215 as representative concentrations. In the presence of 20 nM MCAK, dynamic microtubules could grow, but their catastrophe frequency was significantly increased, and their lifetime was shortened (*Figure 7A and B*). In contrast, 50 nM XMAP215 increased the growth rate more than fourfold (control: 0.8±0.1 µm min⁻¹, n=97 microtubules from 6 assays; 50 nM XMAP215: 3.7±0.3 µm min⁻¹, n=50 microtubules from 3 assays). Notably, XMAP215 alone also caused a mild shortening of microtubule lifetime (*Figure 7A and B*; 50 nM XMAP215: 264.6±12.9 s, n=237 growing events from 3 assays), as compared to that of the control group (323.4±12.6 s of control, n=443 growing events from 6 assays; p>1, the two-tailed Mann-Whitney U test with Bonferroni correction). This is consistent with previous reports suggesting that XMAP215 enhances fluctuations in microtubule end structure (*Farmer et al., 2021*). When both 20 nM MCAK and 50 nM XMAP215 were added, the microtubule growth rate was comparable to that observed with XMAP215 alone (3.9±0.3 µm min⁻¹, n=52 microtubules from 3 assays; p=0.5, the two-tailed Mann-Whitney U test with Bonferroni correction), suggesting that MCAK does not significantly affect XMAP215's function as a polymerase. However, the lifetime of dynamic microtubules (170.3±7.5 s, n=283 growing events from 3 assays) was shorter than that observed with XMAP215 alone (p=2×10⁻⁶, the two-tailed Mann-Whitney U test with Bonferroni correction), while it was similar to that observed with MCAK alone (196.7±6.6 s, n=621 growing events from 6 assays; p>1, the two-tailed Mann-Whitney U test with Bonferroni correction; *Figure 7B*).

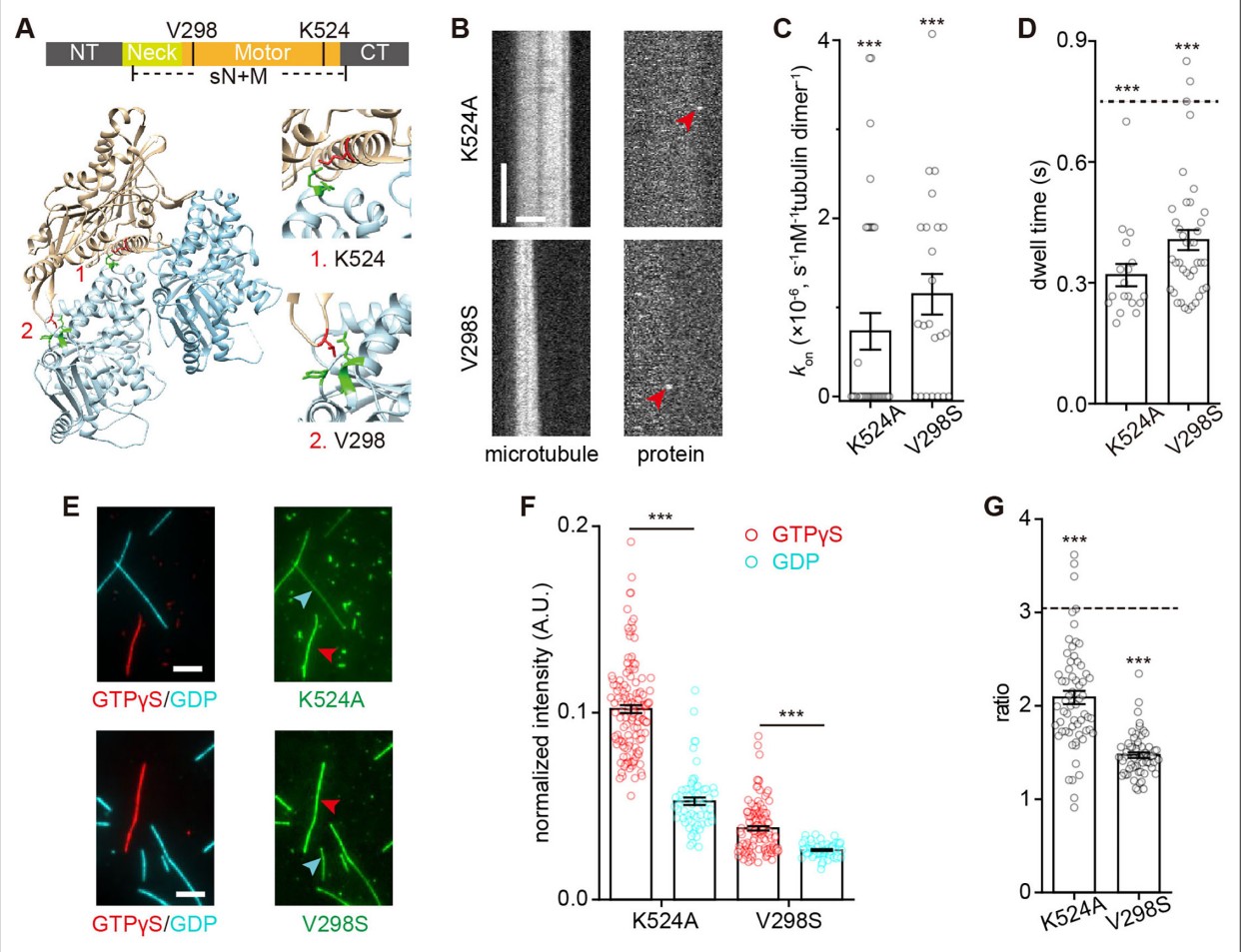

**Figure 6.** The α4 helix and the L2 loop of MCAK contribute to the preferential binding to the EB cap. (**A**) Structural model of the MCAK$^{sN+M}$-tubulin complex (PDB ID: 5MIO; X-ray diffraction). Top: Domain organization of MCAK. Bottom: Enlarged view of α4 helix and L2 loop, regions mediating the interaction between MCAK and tubulin, with the key sites (K524 and V298) highlighted in red. The residues in tubulin that potentially interact with K524 and V298 were highlighted in green. (**B**) Representative kymographs of single-molecule binding events of GFP-MCAK$^{K524A}$ (10 nM) and GFP-MCAK$^{V298S}$ (30 nM) at growing microtubule ends (red, tubulin: 16 μM) in the presence of 1 mM ATP. The end-binding events were indicated by red arrowheads. Scale bars: vertical, 5 s; horizontal, 2 μm. (**C**) Statistical quantification of on-rate ($k_{on-P}$) of GFP-MCAK$^{K524A}$ (34 microtubules from 3 assays) and GFP-MCAK$^{V298S}$ (23 microtubules from 3 assays) at growing microtubule ends in the presence of 1 mM ATP. The data were presented as mean ± SEM and were compared to that of wild-type GFP-MCAK (from ***Figure 1***). (**D**) Statistical quantification of dwell time of GFP-MCAKK$^{524A}$ (18 binding events from 3 assays) and GFP-MCAK$^{V298S}$ (39 binding events from 3 assays) at growing microtubule ends in the presence of 1 mM ATP. The data were presented as mean ± SEM and were compared to that of wild-type GFP-MCAK in the presence of ATP (from ***Figure 1***). The dashed line: the mean dwell time of wild-type GFP-MCAK. (**E**) Representative fluorescence projection images of 5 nM GFP-MCAK$^{K524A}$ and 10 nM GFP-MCAK$^{V298S}$ binding to GTPγS (red arrowhead) versus GDP (cyan arrowhead) microtubules in the presence of 1 mM AMPPNP. Scale bar: 5 μm. (**F**) Statistical quantification of normalized binding intensity of 1 nM GFP-MCAK$^{K524A}$ (134 GTPγS microtubules and 62 GDP microtubules from 3 assays) and GFP-MCAK$^{V298S}$ (124 GTPγS microtubules and 59 GDP microtubules from 3 assays) on different microtubules. All the data were normalized to the binding intensity of wild-type GFP-MCAK at GTPγS microtubules in the AMPPNP condition (from ***Figure 4***). Data were presented as mean ± SEM. (**G**) The binding intensity ratios of GFP-MCAK$^{K524A}$ or GFP-MCAK$^{V298S}$ on GTPγS microtubules to that on GDP microtubules in the presence of 1 mM AMPPNP. GFP-MCAK$^{K524A}$: n=62 GTPγS/GDP binding ratios from 3 assays in AMPPNP state. GFP-MCAK$^{V298S}$: n=59 GTPγS/GDP binding ratios from 3 assays in AMPPNP state. Dashed line: the GTPγS/GDP ratio of wild-type GFP-MCAK. The data were presented as mean ± SEM and were compared to that for wild-type GFP-MCAK (from ***Figure 4***). In panels **C**, **D**, **F**, and **G**, all comparisons were performed using two-tailed Mann-Whitney U test with Bonferroni correction, n.s., no significance; *, p<0.05; **, p<0.01; ***, p<0.001.

The online version of this article includes the following source data and figure supplement(s) for figure 6:

**Source data 1.** Numerical data used to generate ***Figure 6***.

**Figure supplement 1.** GFP-MCAK$^{K525A}$ and GFP-MCAK$^{V298S}$ are depolymerizing-deficient and ATPase activity-reduced mutants.

**Figure supplement 1—source data 1.** Numerical data used to generate ***Figure 6—figure supplement 1***.

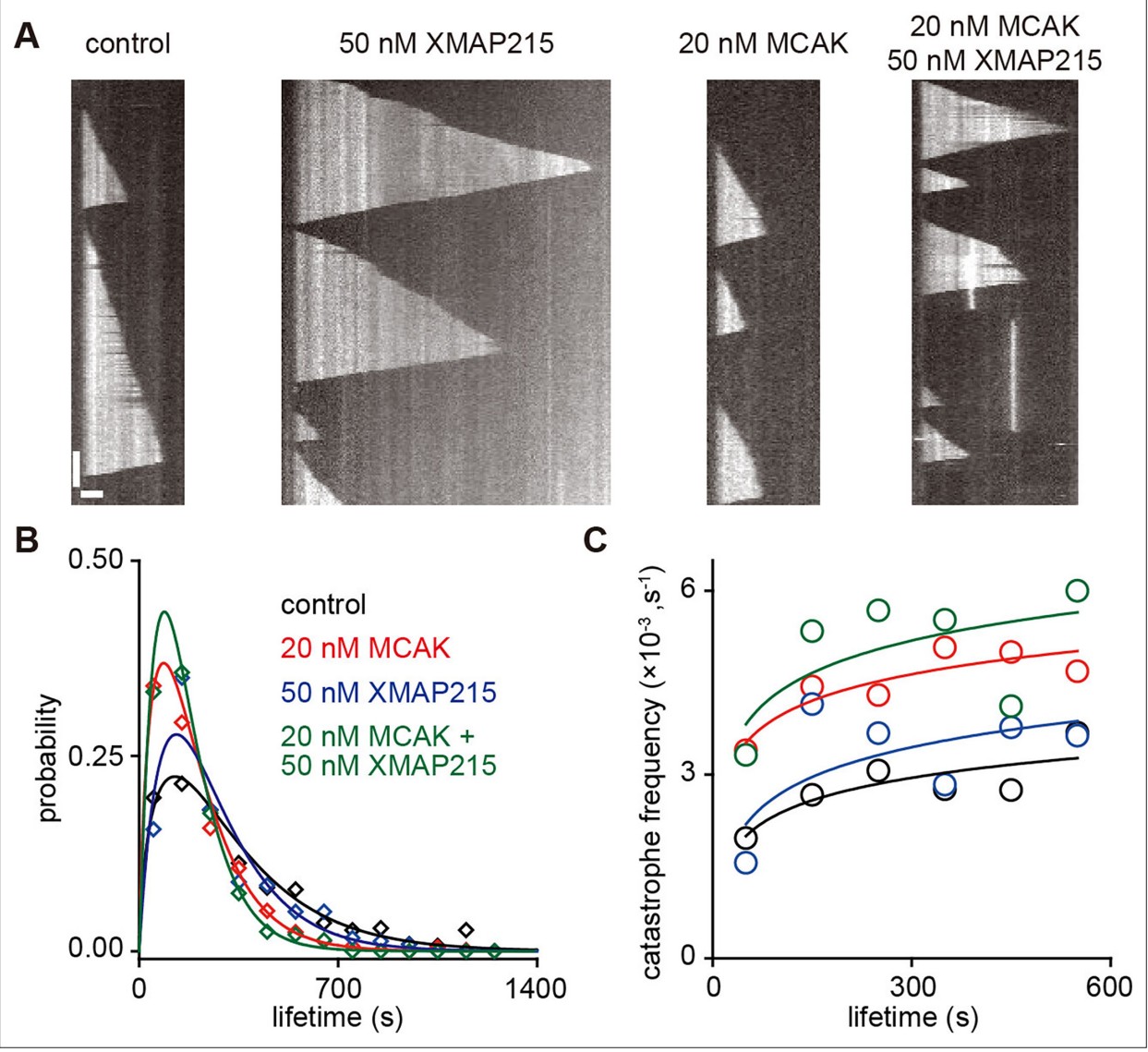

**Figure 7.** Functional specification of MCAK and XMAP215 at growing microtubule ends. (**A**) Representative kymographs of dynamic microtubules (tubulin: 10 μM) under four conditions: control (no MAPs), 20 nM MCAK (1 mM ATP), 50 nM XMAP215 or both (1 mM ATP). Scale bars: vertical, 100 s; horizontal, 2 μm. (**B**) Probability distribution of the microtubule lifetime under the conditions indicated in the panel **A**. Lines represent gamma function fits. Control (black): n=443 microtubules from 7 assays. 20 nM MCAK (red): n=621 microtubules from 7 assays. 50 nM XMAP215 (blue): n=237 microtubules from 3 assays. 20 nM MCAK +50 nM XMAP215 (green): n=283 microtubules from 3 assays. (**C**) The probability-based catastrophe frequency versus microtubule lifetime showing how the likelihood of catastrophe depended on the age of microtubules. Data derived from panel **A**. Control (black): n=443 microtubules from 7 assays. 20 nM MCAK (red): n=621 microtubules from 7 assays. 50 nM XMAP215 (blue): n=237 microtubules from 3 assays. 20 nM MCAK +50 nM XMAP215 (green): n=283 microtubules from 3 assays.

The online version of this article includes the following source data for figure 7:

**Source data 1.** Numerical data used to generate *Figure 7*.

To determine whether this reflects the simple summation of the individual contributions of MCAK and XMAP215 or a synergistic effect, we calculated the probability-based catastrophe frequency for all conditions (*Figure 7C*; *Gardner et al., 2011*). We found that the effects of MCAK and XMAP215 on the probability of microtubule catastrophe were additive, indicating that there was no synergistic effect on catastrophe frequency (*Figure 7C*). This suggests that, at least at this concentration combination, MCAK and XMAP215 independently regulate different parameters of microtubule dynamics, namely catastrophe frequency and growth rate, while simultaneously interacting with microtubule ends.

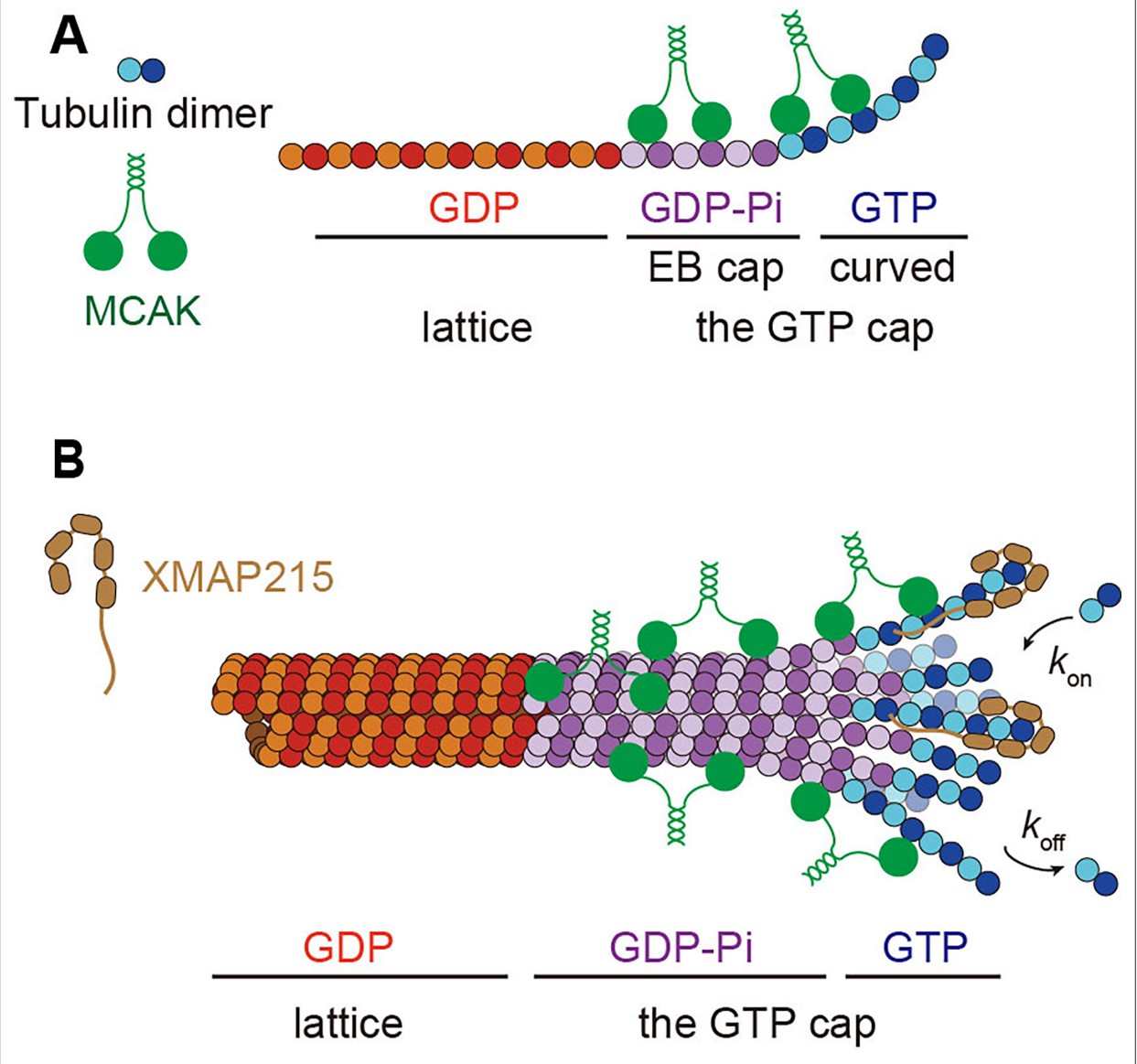

**Figure 8.** Cartoon schematics depicting the working model of MCAK at growing microtubule ends. (**A**) A hypothetical model for the binding of MCAK at growing microtubule ends. (**B**) Cooperation schematics of MCAK and XMAP215 at growing microtubule ends.

## Discussion

This study provides mechanistic insights into the end-binding mechanism of MCAK. The previously established view on the end-binding mechanism of MCAK is based on its specific recognition of curved protofilaments at the distalmost end of growing microtubules (*Asenjo et al., 2013*; *Benoit et al., 2018*; *Moores and Milligan, 2008*; *Mulder et al., 2009*; *Tan et al., 2008*; *Wang et al., 2017*). The findings presented here expand on this model by showing that MCAK also recognizes the nucleotide-dependent feature of growing microtubule ends, more specifically the EB cap where the GDP·Pi-tubulins accumulate (*Figure 8A*). Below, we discuss our key findings and their implications.

### MCAK binds to the EB cap at growing microtubule ends

Our primary discovery is that MCAK preferentially binds to the EB cap, where the GDP·Pi-tubulins accumulate. This conclusion is based on four main lines of evidence. First, localization analysis demonstrates that MCAK not only binds to the distalmost cap but also to the EB cap (*Figure 2B*). Second, MCAK·ADP shows the significantly reduced, but not abolished, binding preference for growing

microtubule ends (*Figure 3*), while their binding preference for curved protofilaments is absent (*Figure 5C*). These different binding behaviors suggest that additional structural features, beyond curved protofilaments, contribute to MCAK's growing end-binding preference. Third, MCAK preferentially binds to GTPγS microtubules over GDP microtubules (*Figure 4*). Since the tubulins in GTPγS microtubules are considered to mimic, at least, some of the key features of the GDP·Pi-tubulins in the EB cap (*Maurer et al., 2011*; *Maurer et al., 2012*), this suggests that MCAK can recognize the molecular features of tubulin dimers within the EB cap. Fourth, previous cryo-EM studies have shown that GTP hydrolysis leads to longitudinal compression of tubulin dimers, increasing the contact area at the interdimer interface (*LaFrance et al., 2022*; *Manka and Moores, 2018*; *Zhang et al., 2015*). Therefore, the interface in GTPγS microtubules is characteristic, being larger than that of a straight GMPCPP-tubulin lattice but smaller than that of a straight GDP-tubulin lattice. We find that the L2 loop contributes to the binding preference to the EB cap, and it was previously reported that the L2 loop of MCAK directly contacts the interdimer interface (*Figure 6A*; *Trofimova et al., 2018*; *Wang et al., 2017*), potentially providing a structural basis for MCAK to distinguish the nucleotide-dependent features of tubulin dimers.

## The end-binding affinity of MCAK

In our experiments, we observed that MCAK exhibits a relatively low end-binding affinity (69 μM), raising the question of its physiological relevance. Several factors may contribute to this observation. First, the in vitro system used may not fully recapitulate physiological conditions. For instance, lowering the salt concentration significantly increases binding affinity (*Helenius et al., 2006*; *Maurer et al., 2011*; *McHugh et al., 2019*). Additionally, while numerous binding events with short durations were detected, we excluded transient interactions from our analysis to facilitate quantification. This likely led to an underestimation of the on-rate and, consequently, the binding affinity. Furthermore, previous studies have shown that retaining a His-tag markedly enhances microtubule binding (*Maurer et al., 2011*; *Zhu et al., 2009*). To minimize interference from purification tags, we ensured complete removal of the His-tag during protein preparation (*Figure 1—figure supplement 1*), which may explain the lower apparent binding affinity in our study compared to others. Finally, a low affinity is not necessarily unexpected. Considering the microtubule end as a receptor with multiple binding sites for MCAK, the overall binding affinity is in the nanomolar range (260 nM). Indeed, only a few MCAK molecules may be sufficient to induce microtubule catastrophe.

A natural follow-up question is the physiological relevance of MCAK's direct binding to microtubule ends compared to the well-established EB1-mediated pathway (*Montenegro Gouveia et al., 2010*; *Honnappa et al., 2009*; *Lee et al., 2008*; *McHugh and Welburn, 2023*). The relative contributions of these two pathways depend on MCAK's end-binding affinity. To estimate their proportional contributions, we employed a simple model (Appendix 1 and *Appendix 1—figure 1*). The modeling results suggest that MCAK's direct end-binding affinity contributes through two distinct mechanisms (Appendix 1). First, MCAK establishes an EB1-independent regulatory pathway. Simultaneously, it enhances the EB1-dependent pathway by increasing the microtubule-end affinity of the MCAK-EB1 complex. Notably, these two mechanisms are not mutually exclusive. Particularly intriguing is that the direct regulatory pathway may account for previously observed experimental phenomena. For example, prior studies have shown that both MCAK and EB1 accumulate at the centromere and centrosome during cell division (*Bahmanyar et al., 2009*; *Fodde et al., 2001*; *Kline-Smith et al., 2004*; *Maney et al., 1998*; *Wordeman and Mitchison, 1995*). However, EB1 neither influences MCAK's localization at these sites nor significantly modulates its local functions (*Domnitz et al., 2012*).

## Implications for the working mechanism of MCAK

We think that our findings add to the current working model of MCAK in several ways. First, the end-binding preference of MCAK for growing microtubule ends is nucleotide state-dependent (*Figure 3*). The present study demonstrates that MCAK exhibits the strongest end-binding preference in the ATP-bound state, suggesting that MCAK·ATP could more effectively distinguish between the end and the lattice. This supports the view that MCAK acts at growing microtubule ends in the ATP-bound state (*Asenjo et al., 2013*; *Benoit et al., 2018*; *Desai et al., 1999*; *Tan et al., 2008*; *Wang et al., 2015*).

Second, the binding preference of MCAK to the GDP·Pi-tubulins provides a mechanism for MCAK remaining bound to the GTP cap after ATP hydrolysis. In a speculative scenario, after the terminal

tubulin dimers are removed, the MCAK motor domain transitions to the ADP state and loses its affinity for curved protofilaments. However, it still preferentially binds to GDP·Pi-tubulins, allowing MCAK to stay within the GTP cap. As ADP is replaced by a new ATP, the motor domain could locally diffuse to explore a new working site for the next round of catalysis, which may facilitate MCAK's processivity (*Cooper et al., 2010*; *Friel and Howard, 2011*; *Helenius et al., 2006*; *Hunter et al., 2003*).

Third, we found that XMAP215 and MCAK independently regulate the growth rate and catastrophe frequency of microtubules, respectively (*Figure 7* and *Figure 8B*). Previous studies have suggested that XMAP215 and MCAK exert antagonistic effects on microtubule assembly (*Barr and Gergely, 2008*; *Kinoshita et al., 2001*; *Moriwaki and Goshima, 2016*; *Tournebize et al., 2000*), possibly through competition for binding sites. Structural studies have indicated overlap in the binding sites of MCAK and XMAP215 on tubulin dimers (*Ayaz et al., 2012*; *Trofimova et al., 2018*), lending support to this hypothesis. However, in the present study, we observed that the binding regions of XMAP215 and MCAK at growing microtubule ends are not entirely overlapping, with MCAK predominantly binding proximally to the binding region of XMAP215 at growing microtubule ends. These findings suggest an alternative mechanism for the combined effects of XMAP215 and MCAK. In this model, MCAK and XMAP215 independently regulate catastrophe frequency and growth rate, respectively, modulating microtubule dynamics through additive effects, rather than through direct competition for binding sites. This model may provide a mechanistic explanation for why ch-TOG (the mammalian homolog of XMAP215) doesn't suppress MCAK activity in mammalian interphase cells (*Holmfeldt et al., 2004*).

## Materials and methods
### Protein expression and purification
#### Protein expression in Sf9 cells
Spodoptera frugiperda (Sf9) cells were utilized for the expression of MCAK, its mutants, and XMAP215 protein, employing a modified Bac-to-Bac baculovirus expression system. The Sf9 cell line (Thermo Fisher Scientific, Cat. # 11496015; RRID:CVCL_0549) was kindly provided by the Protein Facility at Tsinghua University. Cells were maintained in SFM SF (SinoBiological, Cat. # MSF1) at 27°C. Recombinant bacmids were generated by transforming DH10Bac competent cells (Thermo Fisher Scientific, Cat. # 10361012) with the respective protein constructs. For baculovirus production, Sf9 cells were seeded at a density of $1\times10^6$ cells per well in a 12-well tissue culture plate. When cell confluence reached ≥80%, transfection was performed using approximately 1 µg of bacmid DNA and 2 µl of *Trans*IT-Insect Transfection Reagent (Mirus, Cat. # MIR 6100). Following a 48–72 hr incubation post-transfection, the culture supernatant was collected and centrifuged at 2000 × *g* for 5 min to remove cell debris. The resulting supernatant, designated as the P1 viral stock, was stored at 4°C in the dark. To amplify the virus, 50 µl of P1 stock was used to infect 50 ml of Sf9 cells at a density of $0.5\times10^6$ cells/ml. After 72 hr of culture, the P2 viral supernatant was harvested. For protein expression, 1 L of Sf9 culture ($1–2\times10^6$ cells/ml) was infected with 10 ml of P2 baculovirus and incubated for 72 hr at 27°C. Cells were then harvested by centrifugation at 500 × *g* for 15 min and stored at –80°C for subsequent protein purification.

#### MCAK purification
GFP-MCAK, its mutants, and MCAK-RFP (construct schematics shown in *Figure 1—figure supplement 1*) were expressed using a modified BAC-to-BAC protocol. Purification was performed with adaptations based on previously described methods (*Helenius et al., 2006*). Briefly, the Sf9 cells were lysed by Dounce homogenization with ice-cold lysis buffer. The lysate was centrifuged at 40,000 rpm for 60 min at 4°C (XPN-100 ultracentrifuge, Type 45 Ti rotor, Beckman, USA). Cleared supernatant was filtered through a 0.45 µm filter (Pall, Cat. # 66229) and then purified using a cation-exchange column (HiTrap SP-HP, Cat. # 17115101, GE, USA). The protein was eluted using the elution buffer supplemented with a continuous salt gradient (0.15~1.0 M NaCl). Peak fractions were pooled together and then subjected to a Ni-sepharose column (QIAGEN, Cat. # 30210). The column was then washed with the washing buffer I. After the washing step, 3 C Protease (final concentration 10 µg/ml; Thermo Fisher Scientific, Cat. # 88946) was employed to perform on-column his-tag cleavage overnight at 4°C. The successful removal of His-tag was subsequently confirmed using western blot analysis

(*Figure 1—figure supplement 1*). Cleaved protein was further purified through size-exclusion chromatography (Superdex 200 increase 10/300 GL column, Cat. # 28990944, Cytiva, USA) and eluted with the washing buffer II. Purified MCAK aliquots were frozen in liquid $N_2$ and stored at –80°C.

Lysis buffer: 50 mM HEPES pH 7.5, 150 mM NaCl, 5% glycerol, 0.1% Tween 20, 1.5 mM $MgCl_2$, 3 mM EGTA, 1 mM DTT, 0.5 mM Mg-ATP, 10 units/ml Benzonase. Elution buffer: 6.7 mM HEPES pH 7.5, 6.7 mM MES, 6.7 mM sodium acetate, 1.5 mM $MgCl_2$, 10 µM Mg-ATP. Washing buffer I: 50 mM $NaPO_4$ buffer pH 7.5, 300 mM NaCl, 10 mM imidazole, 10% glycerol, 1 mM $MgCl_2$, 10 µM Mg-ATP. Washing buffer II: BRB80, 300 mM KCl, 1 mM DTT, and 10 µM Mg-ATP. BRB80: 80 mM PIPES/KOH pH 6.9, 1 mM $MgCl_2$, 1 mM EGTA.

## EB1 purification

EB1-GFP (construct schematic shown in *Figure 1—figure supplement 1*) was expressed and purified as previously described (*Song et al., 2020*). Briefly, EB1-GFP was expressed in the *BL21 E. coli* strain and lysed by sonication with ice-cold lysis buffer. The lysate was centrifuged at 40,000 rpm for 60 min at 4°C. Cleared supernatant was filtered through a 0.45 µm filter and loaded onto a Ni-sepharose column. The column was then washed using the washing buffer. After the washing step, 3 C Protease (final concentration 10 µg/ml) was employed to perform on-column his-tag cleavage overnight at 4°C. Eluates were desalted into storage buffer using the PD-10 desalting column (GE, Cat. # 17085101). The final purified protein aliquots were frozen in liquid $N_2$ and stored at –80°C.

Lysis buffer: 50 mM $NaPO_4$ buffer pH 7.5, 300 mM NaCl, 10% glycerol, 10 mM imidazole, 1 mM DTT, 0.1% Tween 20. Washing buffer: 50 mM $NaPO_4$ buffer pH 7.5, 300 mM NaCl, 10% glycerol, 30 mM imidazole, 1 mM DTT. Storage buffer: BRB80, 100 mM KCl, 10% glycerol and 1 mM DTT.

## XMAP215 purification

XMAP215-GFP-$his_6$ (construct schematic shown in *Figure 1—figure supplement 1*) was expressed using a modified BAC-to-BAC protocol. The purification was performed as previously described (*Brouhard et al., 2008*). Briefly, the cells were lysed by Dounce homogenization with ice-cold lysis buffer. The lysate was centrifuged at 40,000 rpm for 60 min at 4°C. Supernatant was filtered through a 0.45 µm filter and then purified using a cation-exchange column. The protein was eluted using the elution buffer supplemented with a continuous salt gradient (0.15~1.0 M NaCl). Peak fractions were pooled together and then loaded onto a Ni-sepharose column. The column was then washed with the washing buffer I. The protein was eluted using the washing buffer I supplemented with a continuous imidazole gradient (20–300 mM). Eluted fractions with best purity were pooled together and further purified was further purified through size-exclusion chromatography (Superdex 200 increase 10/300 GL column, Cat. # 28990944, Cytiva, USA) and eluted using the washing buffer II. The final purified XMAP215 was frozen in liquid $N_2$ and stored at –80°C.

Lysis buffer: 50 mM HEPES pH 7.5, 50 mM NaCl, 5% glycerol, 0.1% Tween 20, 1 mM DTT, 10 units/ml Benzonase. Elution buffer: 6.7 mM HEPES pH 7.5, 6.7 mM MES, 6.7 mM sodium acetate. Washing buffer I: 50 mM NaPO4 buffer pH 7.5, 300 mM NaCl, 10 mM imidazole, 10% glycerol, 1 mM $MgCl_2$, 10 µM Mg-ATP. Washing buffer II: BRB80, 150 mM KCl, 1 mM DTT.

## Tubulin preparation

Tubulin was purified from porcine brain tissues through two cycles of temperature-dependent polymerization and depolymerization followed by affinity purification using the TOG-based affinity column, as previously described (*Gell et al., 2010*; *Widlund et al., 2012*). Purified tubulin was labeled with biotin (Thermo Fisher Scientific, Cat. # 20217), TAMRA (Thermo Fisher Scientific, Cat. # C1171) and Alexa Fluor 647 (Thermo Fisher Scientific, Cat. # A20106) using the NHS esters according to the standard protocols (*Gell et al., 2010*).

## Size-exclusion chromatography

To confirm the oligomeric state of the purified proteins, size-exclusion chromatography analysis (Superose 6 Increase 5/150 GL column, Cat. # 29091597, cytiva, USA) was performed using BRB80 supplemented with 50 mM KCl and 1 mM ATP.

## Microtubule depolymerization assay

Microtubules (5% Alexa Fluor 647 labeled and 20% biotin labeled) were polymerized in the presence of GMPCPP (JenaBioscience, Cat. # NU-405L) as previously described (*Song et al., 2020*). The GMPCPP-stabilized microtubules were immobilized on the surface of a cover glass using the biotin-NeutrAvidin protein links (Thermo Fisher Scientific, Cat. # 31000). GFP-MCAK was then added into the flow cell in the imaging buffer. The sample was kept at 35°C using a temperature controller (Tokai Hit, Japan). Images were recorded using a total internal reflection fluorescence microscope (TIRFM; Olympus, Japan) equipped with a 100×1.49 N.A. oil TIRF objective (Olympus, Japan) and an Andor 897 Ultra EMCCD camera (Andor, Belfast, UK). Images were recorded every 5 s with a 100ms exposure. Imaging buffer: BRB80 supplemented with 1 mM ATP, 50 mM KCl, 80 mM D-glucose, 0.4 mg/ml glucose oxidase, 0.2 mg/ml catalase, 0.8 mg/ml casein, 1% β-mercaptoethanol, 0.001% Tween 20.

## Microtubule dynamics assay

Microtubule dynamics assay was performed as previously described (*Song et al., 2020*). Briefly, the immobilized GMPCPP-stabilized microtubules were used as the template for microtubule growth. Tubulin dimers (13% TAMRA labeled) and the protein of interest were then added into the flow cell in the imaging buffer. The sample was kept at 35°C using a temperature controller (Tokai Hit, Japan). Images were recorded using a TIRFM (Olympus, Japan). To record microtubule dynamics, images were recorded every 5 s with a 100ms exposure. To record single molecule binding events, images were recorded every 100ms with a 50ms exposure. Imaging buffer: BRB80 supplemented with 1 mM ATP, 2 mM GTP, 50 mM KCl, 0.15% sodium carboxymethylcellulose, 80 mM D-glucose, 0.4 mg/ml glucose oxidase, 0.2 mg/ml catalase, 0.8 mg/mL casein, 1% β-mercaptoethanol, 0.001% Tween 20.

## Screen for single molecules and the calculation of the binding kinetics

The average fluorescence intensity over the dwell time of each binding event was used to represent the fluorescence signal of this binding event. Subsequently, a probability distribution of the fluorescence intensities of all binding events was constructed and fitted by Gaussian fitting. A binding event was deemed to represent a single molecule if its intensity fell within the range ($\mu \pm 2\sigma$) of the probability distribution. In addition, only binding events with a dwell time exceeding the duration corresponding to 2 pixels (~200ms) were included in the statistical analysis. As a control for the dimeric GFP-MCAK, we used the fluorescence intensity probability distribution of GFP-MCAK$^{sN+M}$, a monomeric variant of MCAK (*Wang et al., 2012*; *Figure 1—figure supplement 1*). Subsequently, the number of single-molecule binding events on the ends and the lattices was counted, respectively. The apparent association rate ($k_{on}$) of MCAK on the end or the lattice was calculated as follows: the number of binding events in the corresponding region was divided by the number of tubulin dimers on the end or the lattice. The dwell time of MCAK binding events on the ends and the lattices under the same condition was pooled together for constructing a probability distribution, respectively. The apparent dissociation rate ($k_{off}$) was calculated by fitting the probability distribution to a single exponential function (*Figure 1—figure supplement 1*).

## The diffusion coefficient of MCAK at the lattice of dynamic microtubules

The diffusion coefficient of MCAK on dynamic microtubule lattices was calculated as previously described (*Helenius et al., 2006*). First, we tracked MCAK's position along its trajectory on dynamic microtubule lattices using the TrackMate plugin in ImageJ (Fiji; *Tinevez et al., 2017*), maintaining a minimum track length of 400ms (100ms/frame). Next, the mean-squared displacement (MSD) was computed from 203 trajectories across three assays using MATLAB. Finally, MSD values were plotted against the time interval, and the diffusion coefficient ($D$) was determined by linear regression analysis ($<x^2>=2\,Dt$) using Origin 8.0 (OriginLab Corporation, USA).

## The flux of MCAK reaching the growing microtubule ends

The flux of MCAK reaching growing microtubule ends was calculated by a one-dimensional lattice-diffusion model, based on Fick's first equation of diffusion (*Helenius et al., 2006*):

$$J = -D \cdot \partial c / \partial x \qquad (1)$$

where $J$ represents the flux of MCAK reaching growing microtubule ends, $c(x, t)$ is the concentration of MCAK on microtubule lattice at position $x$ from the growing microtubule end and at time $t$. When

$$x = 0, J_0 = -D \cdot c_\infty / x_0 \tag{2}$$

where

$$x_0 = \left(D/k_{\text{off}}\right)^{1/2} \tag{3}$$

$$c(x \to \infty) = c_\infty = k_{\text{on}} \cdot C/k_{\text{off}} \tag{4}$$

where $C$ is the concentration of MCAK. Therefore, the flux was calculated by taking the measured parameters ($D$, $k_{\text{on}}$, $C$, and $k_{\text{off}}$) into the equations.

## Localization analysis at growing microtubule ends

The single-molecule localization at growing microtubule ends was conducted as previously described (*Maurer et al., 2014*; *Song et al., 2020*). Briefly, fluorescence calibration of the captured movie was performed using fluorescent beads (T7279, Thermo Fisher Scientific). During the dwell time of a single-molecule binding event, the microtubule lattice was considered as a Gaussian wall and its end as a half-Gaussian in every frame. The peak position of the half-Gaussian was used as a reference to measure the intensity profile of single-molecule binding events frame by frame. A Gaussian function was employed to fit the intensity profile of individual molecules. The position at the peak of the intensity profile was taken as the single molecule's location in every frame (*Figure 1—figure supplement 2*). Subsequently, the average of these positions was taken to represent the single-molecule binding event's position during its dwell time (*Figure 1—figure supplement 2*). This procedure was iterated for all single-molecule binding events to determine their positions during their respective dwell times. Then, the spatial positional distributions of GFP-MCAK and XMAP215-GFP were fitted by Gaussian fitting and the spatial positional distribution of EB1-GFP was fitted to an exponentially modified Gaussian function, from which we calculated the FWHM as an estimation for their length of the binding regions (*Figure 2C*).

## Location precision analysis

The observed image results from the convolution of the fluorescent object with the system's point spread function (PSF). Consequently, the standard deviation of the observed image reflects a combination of measurement error (standard deviation) in determining the object's position and the width of the PSF (*Equation 5*; *Bohner et al., 2016*; *Demchouk et al., 2011*). To characterize the PSF of the imaging system, we used multicolor fluorescent beads (ThermoFisher Scientific, Cat. # T7279) with a 100 nm diameter (*Hirvonen et al., 2009*). When measuring the position of a microtubule end in a single image, we modeled the microtubule lattice as a Gaussian wall and its end as a half-Gaussian. Using this fitting approach, we estimated the measurement error (standard deviation) of the convolved microtubule end positions and deconvolved the system's PSF contribution using *Equation 5*. The mean standard deviation of end positions over the dwell time of a binding event was then used as the measurement error for the microtubule end position. This procedure was iterated for all microtubule ends with single-molecule binding events to construct their measurement errors. Then, the probability distribution of measurement errors was fitted by Gaussian fitting. The standard deviation of the probability distribution was taken as the precision in estimating microtubule end positions (~1.3 nm). The same calculation method was applied to determine the positional precision of other single molecules (~18 nm).

$$\sigma_{\text{I}}^2 = \sigma_{\text{O}}^2 + \sigma_{\text{PSF}}^2 \tag{5}$$

## Microtubule polymerization

GMPCPP, GDP, and GTPγS microtubules were polymerized as previously described (*Manka and Moores, 2018*). Briefly, GMPCPP-microtubules were polymerized using a mixture of 10 µM tubulin, 1 mM GMPCPP (JenaBioscience, Cat. # NU-405L) and 4 mM $MgCl_2$ in BRB80, which was incubated for 2 hours at 37°C. The polymerized microtubules were then pelleted using an Air-Driven Ultracentrifuge

(Beckman, Cat. # 340401), resuspended in BRB80 and stored at 37°C. GDP-microtubules were polymerized using a mixture of 40 μM tubulin, 1 mM GTP (Roche, Cat. # 10106399001), 4 mM MgCl$_2$ and 4% DMSO (Sigma, Cat. # 276855). The mixture was incubated for 30 min at 37°C. The polymerized microtubules were collected using an Air-Driven Ultracentrifuge, resuspended in BRB80 with 20 μM taxol (Cell Signaling Technology, Cat. # 9807) and stored at 37°C. GTPγS microtubules were prepared using two rounds of polymerizations. In the first round, a mixture of 40 μM tubulin, 2 mM GTPγS (Roche, Cat. # 10220647001) and 4 mM MgCl$_2$ in BRB80 was incubated for 5 hr at 37°C. In the second round, 5 μl of the product from the first polymerization was mixed with additional 40 μM tubulin, 2 mM GTPγS and 4 mM MgCl$_2$ to create a final 25 μl reaction mixture. This mixture was incubated overnight at 37°C to obtain long GTPγS microtubules. The polymerized microtubules were collected using an Air-Driven Ultracentrifuge, resuspended in BRB80 with 20 μM taxol and stored at 37°C.

### Microtubule-stimulated ATPase assay

The microtubule-stimulated ATPase activities of MCAK and its mutants were performed by using the Kinesin ATPase End-Point Biochem Kit (Cytoskeleton, Inc, Cat. # BK053). The measurements were based on the malachite green phosphate assay to probe inorganic phosphate generated during the reaction. Wild-type MCAK served as the control, and each type of variant was subjected to multiple replicate measurements.

### Binding of MCAK to different microtubules

The microtubules in different nucleotide states were first immobilized in flow cells. GFP-MCAK or its mutants were then added into the flow cell in the imaging buffer. Images were recorded every 300ms with a 100ms exposure. Imaging buffer: BRB80 supplemented with 1 mM ATP, ADP, or AMPPNP, 20 μM taxol, 50 mM KCl, 80 mM D-glucose, 0.4 mg/ml glucose oxidase, 0.2 mg/ml catalase, 0.8 mg/ml casein, 1% β-mercaptoethanol, 0.001% Tween 20.

### The binding intensity of GFP-MCAK at GMPCPP microtubule ends

To measure the binding intensity of GFP-MCAK at GMPCPP microtubule ends, an intensity profile of GFP-MCAK or the GMPCPP microtubule was plotted along the microtubule. The average binding intensity of GFP-MCAK of three pixels represents the end-binding intensity of GFP-MCAK at the corresponding position of the end, to the lattice from the midpoint of the microtubule intensity attenuation at the microtubule end.

### Gamma fitting of microtubule lifetimes and catastrophe frequency analysis

The lifetime data of dynamic microtubules were fitted to the gamma density function using the dfittool toolbox in MATLAB (Mathworks, USA). The probability-based catastrophe frequency was calculated as the ratio of the number of catastrophe events observed at a certain lifetime to the total number of microtubules that reached this lifetime (*Gardner et al., 2011*).

## Acknowledgements

The authors thank Pengpeng Yu (School of Life Sciences and Technology, TongJi University), Chunguang Wang (School of Life Sciences and Technology, TongJi University), Peng Shi (School of Basic Medical Sciences, Peking University), and Congying Wu (School of Basic Medical Sciences, Peking University) for technical assistance. Special thanks to the light microscopy and protein facility in Tsinghua University. We also acknowledge our funding from the National Natural Sciences Foundation of China (32070704, 32370730), IDG/McGovern Institute for Brain Research (Tsinghua University), Tsinghua-Peking Center for Life Sciences and the State Key Laboratory of Complex, Severe, and Rare Diseases.

## Additional information

### Funding

| Funder | Grant reference number | Author |
|---|---|---|
| National Natural Science Foundation of China | 32070704 | Xin Liang |
| National Natural Science Foundation of China | 32370730 | Xin Liang |

The funders had no role in study design, data collection and interpretation, or the decision to submit the work for publication.

### Author contributions

Wei Chen, Data curation, Formal analysis, Validation, Investigation, Visualization, Methodology, Writing – original draft, Project administration, Writing – review and editing; Yin-Long Song, Investigation, Methodology, Writing – review and editing; Jian-Feng He, Software, Investigation, Methodology, Writing – review and editing; Xin Liang, Conceptualization, Resources, Supervision, Funding acquisition, Methodology, Writing – original draft, Project administration, Writing – review and editing

### Author ORCIDs

Wei Chen ⓘ https://orcid.org/0000-0001-7454-3882
Yin-Long Song ⓘ https://orcid.org/0000-0002-1666-9628
Jian-Feng He ⓘ https://orcid.org/0000-0001-6995-2980
Xin Liang ⓘ https://orcid.org/0000-0001-7915-8094

Reviewer #1 (Public review): https://doi.org/10.7554/eLife.92958.3.sa1
Reviewer #4 (Public review): https://doi.org/10.7554/eLife.92958.3.sa2
Author response https://doi.org/10.7554/eLife.92958.3.sa3

## Additional files

### Supplementary files
MDAR checklist

### Data availability

Source data containing the numerical data used to generate the figures are available in the manuscript. Source data of the original files for the western blot and the gel displayed in Figure 1—figure supplement 1B and J, and PDF file containing original western blot and the gel for Figure 1—figure supplement 1B and J, indicating the relevant bands, are available in the manuscript. The MATLAB script tracking the location of Microtubule-Associated Proteins (MAPs) at growing microtubule ends presented in this paper is openly accessible at GitHub, copy archived at *Liang, 2025*.

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

## Appendix 1

### A simple model for the functional contribution of MCAK's direct end-binding mechanism

Here, we presented a simple model to calculate the relative contribution of the direct and EB-dependent end-binding of MCAK (**Appendix 1—figure 1**).

**Appendix 1—figure 1.** A simple model for the end-binding of MCAK and EB1. MCAK can bind to growing microtubule ends through both the direct (left) and EB-dependent (right) pathways. The dissociation constants were $K_0$, $K_1$, $K_2$, and $K_3$, respectively. MTE: growing microtubule end.

Based on the model, we had the dissociation constants:

$$K_0 = \frac{[MCAK] \cdot [EB1]}{[MCAK \cdot EB1]} \tag{6}$$

$$K_1 = \frac{[MCAK] \cdot [MTE]}{[MCAK - MTE]} \tag{7}$$

$$K_2 = \frac{[EB1] \cdot [MTE]}{[EB1 - MTE]} \tag{8}$$

$$K_3 = \frac{[MCAK \cdot EB1] \cdot [MTE]}{[MCAK \cdot EB1 - MTE]} \tag{9}$$

Then, the relative contribution of the direct and EB-dependent end-binding of MCAK can be expressed as $\alpha$:

$$\alpha = \frac{[MCAK - MTE]}{[MCAK \cdot EB1 - MTE]} = \frac{K_3 \cdot [MCAK]}{K_1 \cdot [MCAK \cdot EB1]} = \frac{K_3 \cdot K_0}{K_1 \cdot [EB1]} \tag{10}$$

Here, we considered two scenarios. In the cytoplasm, both EB1 and MCAK undergo free diffusion and can associate with each other without restrictions. The relative concentrations of MCAK and EB1 are critical parameters, but they may vary across different cell types and remain unknown. We also considered the second scenario in which MCAK is locally enriched at specific cellular localizations through an EB-independent mechanism. For example, EB1 does not affect the localization of MCAK at centromere and centrosome, nor does EB1 significantly affect the function of MCAK there (**Domnitz et al., 2012**). Here, we assumed that the local concentration of anchored-state MCAK is relatively high, and EB1 remains diffusive and its concentration is nearly constant, as it is continuously replenished in the local space from the vast cytoplasmic pool. In both cases, the ratio of $K_3$ to $K_1$ emerges as a key determinant.

$K_3$ is the end-binding affinity of the MCAK·EB1 complex. Intuitively, it depends on the respective microtubule-binding affinities of MCAK and EB1, as well as the cooperativity, if any, of their microtubule-binding behaviors. $K_3$ can be expressed as:

$$\frac{1}{K_3} = a\frac{1}{K_1} + b\frac{1}{K_2} \tag{11}$$

$$K_3 = \frac{K_2 \cdot K_1}{aK_2 + bK_1} \tag{12}$$

where $a$ and $b$ represent the weighting factors of binding sites or cooperativity factors of the binding behaviors. Therefore, $K_3$ shows a positively correlated, monotonically increasing dependence on $K_1$, indicating that the increase in the end-binding affinity of MCAK contributes to that of the MCAK·EB1 complex. Therefore, we think that MCAK's functional impact at microtubule ends derives not only from its intrinsic end-binding capacity, but also its ability to strengthen the EB1-mediated end association pathway.

In the simplest case, the formation of the MCAK·EB1-MTE complex arises from the binding of either MCAK or EB1 to microtubule ends, and the binding behaviors for MCAK and EB1 are independent ($a$=1; $b$=1). Consequently, $K_3$ can be expressed as:

$$K_3 = \frac{K_2 \cdot K_1}{K_2 + K_1} \tag{13}$$

if $K_1 \ll K_2$, then

$$K_3 \approx K_1 \tag{14}$$

if $K_1 \gg K_2$, then

$$K_3 \approx K_2 \tag{15}$$

If $K_1 \approx K_2$, then

$$K_3 \approx \frac{K_1}{2} \tag{16}$$

In our experiments, we measured the dissociation constants of MCAK to growing microtubule ends is 69 µM ($K_1$). We also performed similar experiments with EB1 and found that EB1 showed the dissociation constant of 722 µM ($K_2$) for growing microtubule ends (*Appendix 1—figure 2*), similar to the value reported in our previous report (*Song et al., 2020*). Therefore, substituting *Equation 14* into *Equation 10*, we obtained

$$\alpha \approx \frac{K_0}{[EB1]} \tag{17}$$

Here, if we assume that the cytoplasmic concentration of EB1 is twice the value of $K_0$, then $\alpha$=0.5, indicating that 50% of MCAK binds to microtubule ends via the direct binding pathway; even if the EB1 concentration reaches ten times the value of $K_0$, 10% of MCAK still utilizes the direct binding pathway. Overall, as the EB1 concentration increases relative to $K_0$, α decreases, reflecting a decline in the proportion of MCAK that associates with microtubule ends through the direct binding mechanism.

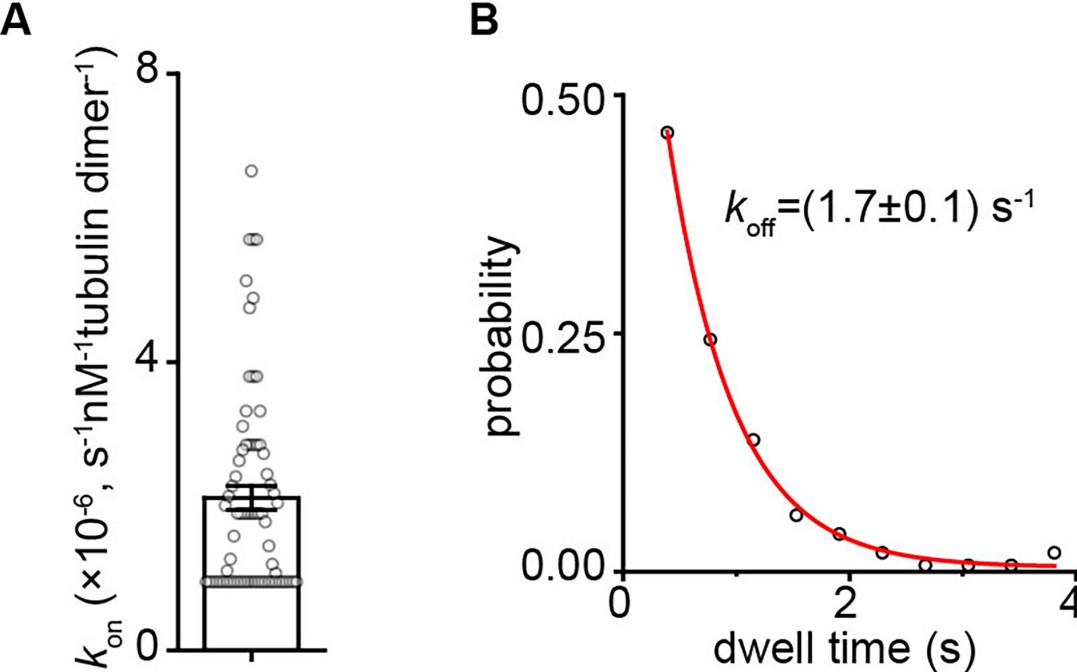

**Appendix 1—figure 2.** The binding kinetics of single-molecule EB1-GFP binding to growing microtubule ends. (**A**) Statistical quantification of on-rate ($k_{on}$) of EB1-GFP's binding to the plus end of dynamic microtubules (data calculated from *Figure 2*, n=71 microtubules from 3 assays). (**B**) The apparent off-rate ($k_{off}$) of EB1-GFP at growing microtubule ends (data calculated from *Figure 2*, n=153 binding events from 3 assays). $k_{off}$ was calculated by fitting the dwell time of individual EB1-GFP binding events to a single exponential function.

The online version of this article includes the following source data for appendix 1—figure 2:

**Appendix 1—figure 2—source data 1.** Numerical data used to generate *Appendix 1—figure 2*.

