## [Editor Report · eLife Assessment]

This work presents **valuable** new information on the microtubule-binding mode of the microtubule kinesin-13, MCAK, the authors use quantitative single-molecule studies to propose that MCAK preferentially binds to a GDP-Pi-tubulin portion of the microtubule end. However, the evidence provided to support this claim remains **incomplete** and would benefit from more rigorous methodology particularly the diffraction limited experiments do not provide sufficient spatial resolution to support the authors' conclusions. In addition, a more through discussion of the existing literature would further strengthen the manuscript.

---

## [Referee Report · Reviewer #1 (Public review)]

The authors responded to multiple criticisms with additional data and more detailed statistics, in some instances improving the quality of the work. However, I had difficulty understanding some of the authors' responses. The logic was not always apparent, the writing was occasionally confusing or would benefit from more careful wording, and some of the provided responses were superficial or raised new concerns. In some cases, the underlying data needed to support their responses were not shown. Thus, the current version of the manuscript does not sufficiently resolve the following critical issues raised by myself and other reviewers.

(1) A clear new insight into a physiological process or cellular behavior remains lacking. The study largely confirms prior observations of MCAK binding to both the microtubule wall and end. However, it is still unclear whether direct binding to the tip-as opposed to accumulation via wall diffusion or interaction with other tip-binding proteins-is a significant mechanism.

(2) The newly revealed adenosine-nucleotide-dependent binding preferences do not help clarify MCAK's catalytic function or its mechanisms of tip recognition. Consequently, the final summary figure remains speculative and is not convincingly supported by the data. It is also unclear what exactly is meant by the "working model" (figure title), or by the claim of "a simple rule of how the end-binding regulators coordinate their activities" (abstract).

(3) As noted in my previous review, the effects of adding different adenosine nucleotides on MCAK binding to microtubules are much more pronounced than the differences in MCAK binding to tubulin with various guanosine-containing nucleotides, or to lattice versus tip (e.g., Fig. 5E). Therefore, the manuscript title-"MCAK recognizes the nucleotide-dependent feature at growing microtubule ends"-does not do justice to the scale of these effects.

(4) The title implies that MCAK selectively recognizes a feature determined by the tubulin-bound guanosine nucleotide. However, the authors frequently claim that MCAK binds to the "entire GTP cap." It appears that they exclude structural protrusions from their definition of the cap, which is debatable. Even using their definition, the conclusion that MCAK recognizes a specific "nucleotide-dependent feature" seems inconsistent with the claim that it binds uniformly across the cap. These distinctions were not made clear.

(5) Some important technical details are still absent. For example, when reading the authors' response to another reviewer's question, I could not find an explanation of how the kon values for end and wall binding were calculated. These calculations clearly require assumptions, e.g. about the number of binding sites, but these details are not described. In addition, the binding data are expressed in units per tubulin dimer, which are non-standard and make comparisons to other published results difficult. There are other instances where more technical detail would be desirable, but they are too numerous to list here.

(6) Several aspects of data presentation as graphs will make it difficult for other researchers to analyze or interpret the findings. Numerical Excel-style data sheets should be provided for all measurements, including raw data-not just the ratios or derived values shown in plots. Other, more significant issues include use of mean values for non-Gaussian distributions (e.g., dwell times); binding affinities inferred from single-concentration measurements, often under varying conditions (e.g., Figs. 3C, 4); and absence of side-by-side plotted controls (e.g., Fig. 6).

(7) While the authors have added some quantitative values and descriptive detail, the manuscript still lacks a critical comparison of their findings with existing literature. This weakens the impact of the study and limits the reader's ability to place the results in a broader context.

---

## [Referee Report · Reviewer #4 (Public review)]

The revised manuscript from Chen et al. implements many of the changes requested by the 3 reviewers of the initial submission. These changes are well-described in the corresponding Response to Reviews document. Of course, not every request from the reviewers was addressed, and the following major concerns remain:

(1) The authors argue that MCAK binds to the same region as EB proteins, which they refer to as the "EB cap". Reviewers asked for experiments that would increase the size of the EB cap to create "comets" (e.g. by increasing the microtubule growth rate); the prediction is that the MCAK signal should increase in size as well. The authors declined to pursue these experiments. As a result, the EB signals and MCAK signals are diffraction-limited spots, as opposed to the predicted exponential decay signals characteristic of EB comets. The various diffraction-limited spots are then aligned with the diffraction-limited signal of the microtubule end. These alignments and sub-pixel comparisons are technically challenging. The revised manuscript does not go far enough to provide compelling evidence that all technical challenges were overcome. Thus, while the authors can safely conclude that MCAK, EBs, and the microtubule end do occupy the same diffraction-limited spot, more precise conclusions are not supported.

(2) The reviewers criticized the initial manuscript for neglecting key references, particularly Kinoshita et al., Science 2001. Indeed, I cannot fathom writing a manuscript about MCAK and XMAP215 without putting a citation to such a landmark paper front and center. The authors have responded by including more discussion of the relevant literature (and citing Kinoshita et al.). However, the revised manuscript is often still cursory in giving credit where credit is due, contextualizing the new data, and generally engaging with the scholarship on MCAK.

(3) The data presented does not include a simple measurement of the impact of MCAK on the catastrophe frequency of microtubules. The authors explain this absence by pointing out that their movies are short (5 min) and high frame rate (10 fps). While I understand that such imaging parameters are necessary to capture single molecule end-binding events, I do not understand why a separate set of experiments could not be performed. This type of "positive control" is often missing, as pointed out by the 3 reviewers.

(4) Salt conditions, protein concentrations, and other key experimental parameters are not varied, even when varying them would provide excellent tests of the authors' hypotheses.

In summary, the revised manuscript is improved in many ways, but the interested reader should look carefully at the previous reviews and compare the measurements presented here with those of other labs.

---

## [Author Response]

The following is the authors’ response to the original reviews.

**Reviewer #1 (Public Review):**
Major concerns:(1) Is the direct binding of MCAK to the microtubule cap important for its in vivo function?a.The authors claim that their "study provides mechanistic insights into understanding the end-binding mechanism of MCAK". I respectfully disagree. My concern is that the paper offers limited insights into the physiological significance of direct end-binding for MCAK activity, even in vitro. The authors estimate that in the absence of other proteins in vitro, ~95% of MCAK molecules arrive at the tip by direct binding in the presence of ~ physiological ATP concentration (1 mM). In cells, however, the major end-binding pathway may be mediated by EB, with the direct binding pathway contributing little to none. This is a reasonable concern because the apparent dissociation constant measured by the authors shows that MCAK binding to microtubules in the presence of ATP is very weak (69 uM). This concern should be addressed by (1) calculating relative contributions of direct and EB-dependent pathways based on the affinities measured in this and other published papers and estimated intracellular concentrations. Although there are many unknowns about these interactions in cells, a modeling-based analysis may be revealing. (2) the recapitulation of these pathways using purifying proteins in vitro is also feasible. Ideally, some direct evidence should be provided, e.g. based on MCAK function-separating mutants (GDP-Pi tubulin binding vs. catalytic activity at the curled protofilaments) that contribution from the direct binding of MCAK to microtubule cap in EB presence is significant.

We thank the reviewer for the thoughtful comments.

(1) We think that the end-binding affinity of MCAK makes a significant contribution for its cellular functions. To elucidate this concept, we now use a simple model shown in Supplementary Appendix-2 (see pages 49-51, lines 1246-1316). In this model, we simplified MCAK and EB1 binding to microtubule ends by considering only these two proteins while neglecting other factors (e.g. XMAP215). Specifically, we considered two scenarios: one in which both proteins freely diffuse in the cytoplasm and another where MCAK is localized to specific cellular structures, such as the centrosome or centromere. Based on the modeling results, we argue that MCAK's functional impact at microtubule ends derives both from its intrinsic end-binding capacity and its ability to strengthen the EB1-mediated end association pathway.

(2) We agree with the reviewer that MCAK exhibiting a lower end-binding affinity (69 µM) is indeed intriguing, as one might intuitively expect a stronger affinity, e.g. in the nanomolar range. Several factors may contribute to this observation. First, this could be partly due to the in vitro system employed, which may not perfectly replicate in vivo conditions, especially when considering cellular processes quantitatively. Variations in medium composition can significantly influence the binding state. For example, reducing salt concentration leads to a marked increase in MCAK’s binding affinity (Helenius et al., 2006; Maurer et al., 2011; McHugh et al., 2019). Additionally, while numerous binding events with short durations were detected, we excluded transient interactions from our analysis to facilitate quantification. This likely leads to an underestimation of the on-rate and, consequently, the binding affinity. Moreover, to minimize the interference of purification tags (His-tag), we ensured their complete removal during protein sample preparation. Previous studies reported that retaining the His-tag of MAPs affects the binding affinity to microtubules (Maurer et al., 2011; Zhu et al., 2009). Finally, a low affinity is not necessarily unexpected. Considering the microtubule end as a receptor with multiple binding sites for MCAK, the overall binding affinity is in the nanomolar range (260 nM). This does not necessarily contradict MCAK being a microtubule dynamics regulator as only a few MCAK molecules may suffice to induce microtubule catastrophe (as discussed on page 13, lines 408-441).

(3) Ideally, we would search for mutants that specifically interfere with the binding of GDP-Pi-tubulin or the curled protofilaments. However, the mutant we tested significantly impacts the overall affinity of MCAK to microtubules (both end and lattice), making it challenging to isolate and discuss the function of MCAK with respect to the binding to GDP-Pi-tubulin alone. Additionally, we also think that the GDP-Pi-tubulin in the EB cap and the tubulin in the curved protofilaments may share structural similarities. For instance, the tubulin dimers in both states may be less compact compared to those in the lattice, which could explain why MCAK recognizes both simultaneously (Manka and Moores, 2018). However, this remains a conjecture, as there is currently no direct evidence to support it.

b. As mentioned in the Discussion, preferential MCAK binding to tubulins near the MT tip may enhance MCAK targeting of terminal tubulins AFTER the MCAK has been "delivered" to the distal cap via the EB-dependent mechanism. This is a different targeting mechanism than the direct MCAK-binding. However, the measured binding affinity between MCAK and GMPCPP tubulins is so weak (69 uM), that this effect is also unlikely to have any impact because the binding events between MCAK and microtubule should be extremely rare. Without hard evidence, the arguments for this enhancement are very speculative.

Please see our response to the comment No. 1. Additionally, we have revised our discussion to discuss the end-binding affinity of MCAK as well as its physiological relevance (please see page 13, lines 408-441; and see Supplementary Appendix-2 in pages 49-51, lines 1246-1316).

(2) The authors do not provide sufficient justification and explanation for their investigation of the effects of different nucleotides in MCAK binding affinity. A clear summary of the nucleotide-dependent function of MCAK (introduction with references to prior affinity measurements and corresponding MCAK affinities), the justifications for this investigation, and what has been learned from using different nucleotides (discussion) should be provided. My take on these results is that by far the strongest effect on microtubule wall and tip binding is achieved by adding any adenosine, whereas differences between different nucleotides are relatively minor. Was this expected? What can be learned from the apparent similarity between ATP and AMPPNP effects in some assays (Fig 1E, 4C, etc) but not others (Fig 1D,F, etc)?

We thank the reviewer for this suggestion. We have revised the manuscript accordingly, and below are the main points of our response

(1) The experiment investigating the effects of different nucleotides on MCAK binding affinity was inspired by the previous studies demonstrating that kinesin-13 interactions with microtubules are highly dependent on their adenosine-bound states. For example, kinesin-13s tightly bind microtubules and prefer to form protofilament curls or rings with tubulin in the AMPPNP state, whereas kinesin-13s are considered to move along the microtubule lattice via one-dimensional diffusion in the ADP·Pi state (Asenjo et al., 2013; Benoit et al., 2018; Friel and Howard, 2011; Helenius et al., 2006). Based on these observations, we wondered whether MCAK's adenosine-bound states might similarly affect its binding preference for growing microtubule ends. We have made the motivation clear in the revised manuscript (please see page 7, lines 199-209).

(2) Our main finding regarding the effects of nucleotides is that MCAK shows differential end-binding affinity and preference based on its nucleotide state. First, MCAK shows the greatest preference for growing microtubule ends in the ATP state, supporting the idea that diffusive MCAK (MCAK·ATP) can directly bind to growing microtubule ends. Second, MCAK·ATP also demonstrates a binding preference for GTPγS microtubules and the ends of GMPCPP microtubules. The similar trends in binding preference suggest that the affinity for GDP·Pi-tubulin and GTP-tubulin likely underpins MCAK’s preference for growing microtubule ends. To clarify these points, we have added further discussions in the manuscript (please see page 8, lines 230-233; page9, lines 258-270 and pages 13-14, lines 443-458).

(3) It is not clear why the authors decided to use these specific mutant MCAK proteins to advance their arguments about the importance of direct tip binding. Both mutants are enzymatically inactive. Both show roughly similar tip interactions, with some (minor) differences. Without a clear understanding of what these mutants represent, the provided interpretations of the corresponding results are not convincing.

We thank the reviewer for this comment. In the revised manuscript, we no longer draw conclusions about the importance of end-binding based on the mutant data. Instead, we think that the mutant data provide insights into the structural basis of the end-binding preference. Therefore, we have rewritten the results in this section to more accurately reflect these findings (please see page 10, lines 295-327).

(4) GMPCPP microtubules are used in the current study to represent normal dynamic microtubule ends, based on some published studies. However, there is no consensus in the field regarding the structure of growing vs. GMPCPP-stabilized microtubule ends, which additionally may be sensitive to specific experimental conditions (buffers, temperature, age of microtubules, etc). To strengthen the authors' argument, Taxol-stabilized microtubules should be used as a control to test if the effects are specific. Additionally, the authors should consider the possibility that stronger MCAK binding to the ends of different types of microtubules may reflect MCAK-dependent depolymerization events on a very small scale (several tubulin rows). These nano-scale changes to tubulins and the microtubule end may lead to the accumulation of small tubulin-MCAK aggregates, as is seen with other MAPs and slowly depolymerizing microtubules. These effects for MCAK may also depend on specific nucleotides, further complicating the interpretation. This possibility should be addressed because it provides a different interpretation than presented in the manuscript.

Regarding the two points raised here, our thoughts are as following

(1) The end of GMPCPP-stabilized microtubules differs from that of growing microtubules, with the most obvious known difference being the absence of the region enriched in GDP-Pi-tubulin. We consider the end of GMPCPP microtubules as an analogue of the distal tip of growing microtubules, based on two key features: (1) curled protofilaments and (2) GMPCPP-tubulin, a close analogue of GTP-tubulin. Notably, both features are present at the ends of both GMPCPP-stabilized and growing microtubules. Moreover, we agree with the suggestion to use taxol-stabilized microtubules as a control. This would eliminate the second feature (absence of GTP-tubulin), allowing us to isolate the effect of the first feature. Therefore, we conducted this experiment, and our data showed that MCAK exhibits only a mild binding preference for the ends of taxol-stabilized microtubules, which is much less pronounced than for the ends of GMPCPP microtubules. This observation supports the idea that GMPCPP-stabilized ends closely resemble the growing ends of microtubules.

(2) The reviewer suggested that stronger MCAK binding to the ends of different types of microtubules might reflect MCAK-dependent depolymerization events on a very small scale. This is an insightful possibility, which we had overlooked in the original manuscript. Fortunately, we performed the experiments at the single-molecule concentrations. Upon reviewing the raw data, we found that under ATP conditions, the binding events of MCAK were not cumulative (see Fig. X1 below) and showed no evidence of local accumulation of MCAK-tubulin aggregates.

**Author response image 1. sa3fig1:** The representative kymograph showing GFP-MCAK binding at the ends and lattice of GMPCPP microtubules in the presence of 1 mM ATP (10 nM GFP-MCAK), which corresponded to Figure 5A. The arrow: the end-binding of MCAK. Vertical bar: 1 s; horizontal bar: 2 mm.

(5) It would be helpful if the authors provided microtubule polymerization rates and catastrophe frequencies for assays with dynamic microtubules and MCAK in the presence of different nucleotides. The video recordings of microtubules under these conditions are already available to the authors, so it should not be difficult to provide these quantifications. They may reveal that microtubule ends are different (or not) under the examined conditions. It would also help to increase the overall credibility of this study by providing data that are easy to compare between different labs.

We thank the reviewer for this suggestion. In the revised manuscript, we have provided data on the growth rates, which are similar across the different nucleotide states (Fig. s1). However, due to the short duration of our recordings (usually 5 minutes, but with a high frame rate, 10 fps), we did not observe many catastrophe events, which prevented us from quantifying catastrophe frequency using the current dataset. Since we measured the binding kinetics of MCAK during the growing phase of microtubules, the similar growth rates and microtubule end morphologies suggest that the microtubule ends are comparable across the different conditions.

**Reviewer #1 (Recommendations For The Authors):**
a. Please provide more details about how the microtubule-bound molecules were selected for analysis (include a description of scripts, selection criteria, and filters, if any). Fig 1A arrows do not provide sufficient information.

We first measured the fluorescence intensity of each binding event. A probability distribution of these intensities was then constructed and fitted with a Gaussian function. A binding event was considered to correspond to a single molecule if its intensity fell within μ±2σ of the distribution. The details of the single-molecule screening process are now provided in the revised manuscript (see page17, lines 574-583).

b. Evidence that MCAK is dimeric in solution should be provided (gel filtration results, controls for Figs1A - bleaching, or comparison with single GFP fluorophore).

In the revised manuscript, we provide the gel filtration results of purified MCAK and other proteins used in this study. The elution volume of the peak for GFP-MCAK corresponded to a molecular weight range between 120 kDa (EB1-GFP dimer) and 260 kDa (XMAP215-GFP-his6), suggesting that GFP-MCAK exists as a dimer (~220 kDa) under experimental condition (please see Fig.s1 and page 5, lines 104-105). In addition, we also measured the fluorescence intensity of both MCAK^sN+M^ and MCAK. MCAK^sN+M^ is a monomeric mutant that contains the neck domain and motor domain (Wang et al., 2012). The average intensity of MCAK^sN+M^ is 196 A.U., about 65% of that of MCAK (300 A.U.). These two measurements suggest that the purified MCAK used in this study exists dimers (see Fig. s1).

c. Evidence that MCAK on microtubules represents single molecules should be provided (distribution of GFP brightness with controls - GFP imaged under identical conditions). Since assay buffers include detergent, which is not desirable, all controls should be done using the same assay conditions. The authors should rule out that their main results are detergent-sensitive.

(1) Regarding if MCAK on microtubules represent single molecules: please refer to our responses to the two points above.

(2) To rule out the effect of tween-20 (0.0001%, v/v), we performed additional control experiments. The results showed that it has no significant effect on microtubule-binding affinity of MCAK (see Figure below).

**Author response image 2. sa3fig2:** Tween-20 (0.0001%, v/v) has no significant effect on microtubule-binding affinity of MCAK. (A) The representative projection images of GFP-MCAK (5 nM) binding to taxol-stabled GDP microtubules in the presence of 1 mM AMPPNP with or without tween-20. The upper panel showed the results of the control experiments performed without MCAK. Scale bar: 5 mm. (B) Statistical quantification of the binding intensity of GFP-MCAK binding to GDP microtubules with or without tween-20 (53 microtubules from 3 assays and 70 microtubules from 3 assays, respectively). Data were presented as mean ± SEM. Statistical comparisons were performed using the two-tailed Mann-Whitney U test with Bonferroni correction, n.s., no significance.

d. How did the authors plot single-molecule intensity distributions? I am confused as to why the intensity distribution for single molecules in Fig 1D and 2A looks so perfectly smooth, non-pixelated, and broader than expected for GFP wavelength. Please provide unprocessed original distributions, pixel size, and more details about how the distributions were processed.

In the revised manuscript, we provided unprocessed original data in Fig. 1B and Fig. 2A. We thank the reviewer for pointing out this problem.

e. Many quantifications are based on a limited number of microtubules and the number of molecules is not provided, starting from Fig 1D and down. Please provide detailed statistics and explain what is plotted (mean with SEM?) on each graph.

We performed a thorough inspection of the manuscript and corrected the identified issues.

f. Plots with averaged data should be supplemented with error bars and N should be provided in the legend. E.g. Fig 1C - average position of MT and peak positions.

We agree with the reviewer. In the revised manuscript, we have made the changes accordingly (e.g. Fig. 2C).

g. Detailed information should be provided about protein constructs used in this work including all tags. The use of truncated proteins or charged/bulky tags can modify protein-microtubule interactions.

We agree with the reviewer. In the revised manuscript, we provide the information of all constructs (see Fig. s1 and the related descriptions in Methods, pages 15-16, lines 476-534).

h. Line 515: We estimated that the accuracy of microtubule end tracking was ~6 nm by measuring the standard error of the distribution of the estimated error in the microtubule end position. - evidence should be provided using the conditions of this study, not the reference to the prior work by others.i. Line 520: We estimated that the accuracy of the measured position was ~2 nm by measuring the standard error of the fitting peak location". Please provide evidence.

Point h-i: we now provide detailed descriptions of how to estimate tracking and measurement accuracy and error in our work. Please see pages 18-19, lines 626-645.

j. Kymographs in Fig 5G are barely visible. Please provide single-channel greyscale images. What are the dim molecules diffusing on this microtubule?

We have incorporated the changes suggested by the reviewer. We think that some of the dim signals may result from stochastic background noise, while others likely represent transient bindings of MCAK. The exposure time in our experiments was approximately 0.05 seconds; if the binding duration were shorter than this, the signal would be lower (i.e. the “dim” signals). It is important to note that in this study, we selected binding events lasting at least 2 consecutive frames, meaning transient binding events were not included. This point has been clarified in the Methods section (see page17, lines 573-583).

k. Please provide a methods description for Fig 6. Did the buffer include 1 mM ATP? The presence of ATP would make these conditions more physiological. ATP concentration should be stated clearly in the main text or figure legend.

The buffer contains ATP. In the revised manuscript, we have provided the methods for the experiments of microtubule dynamics assay, as well as the analysis of microtubule lifetimes and catastrophe frequency (see page 17, lines 561-572 and page 20, lines 685-690).

l. Line 104: experiment was performed in BRB80 supplemented with 50 mM KCl and 1 mM ATP, providing a nearly physiological ion strength. Please provide a reference or add your calculations in Methods.

We have provided references on page 5, lines 101-104 of our manuscript.

m. What was the MCAK concentration in Figure 4? Did the microtubule shorten under any of these conditions?

In these experiments, we used a very low concentration of MCAK and taxol-stabilized microtubules, so there’s no microtubule shortening observed here. ATP: 10 nM GFP-MCAK; AMPPNP: 1 nM GFP-MCAK; ADP: 10 nM GFP-MCAK; APO state: 0.1 nM GFP-MCAK.

Other criticism:Text improvements are recommended in the Discussion. For example, line 348: Fourth, the loss of the binding preference.. suggests that the binding preference .. is required for the optimal .. preference.

We thank the reviewer for pointing out this. In the revised manuscript, we conducted a thorough revision and review of the text.

**Reviewer #2 (Public Review):**
Summary:In this manuscript, Chen et al. investigate the localization of microtubule kinesin-13 MCAK to the microtubule ends. MCAK is a prominent microtubule depolymerase whose molecular mechanisms of action have been extensively studied by a number of labs over the last ~twenty years. Here, the authors use single-molecule approaches to investigate the precise localization of MCAK on growing microtubules and conclude that MCAK preferentially binds to a GDP-Pi-tubulin portion of the microtubule end. The conclusions are speculative and not well substantiated by the data, making the impact of the study in its current form rather limited. Specifically, greater effort should be made to define the region of MCAK binding on microtubule ends, as well as its structural characteristics. Given that MCAK has been previously shown to effectively tip-track growing microtubule ends through an established interaction with EB proteins, the physiological relevance of the present study is unclear. Finally, the manuscript does not cite or properly discuss a number of relevant literature references, the results of which should be directly compared and contrasted to those presented here.

We thank the reviewer for the comments. As these suggestions are more thoroughly expressed in the following comments for authors, we will provide the responses in the corresponding sections, as shown below.

**Reviewer #2 (Recommendations For The Authors):**
Significant concerns:(1) Establishing the precise localization of MCAK wrt microtubule end is highly non-trivial. More details should be provided, including substantial supplementary data. In particular, the authors claim ~6 nm accuracy in microtubule end positioning - this should be substantiated by data showing individual overlaid microtubule end intensity profiles as well as fits with standard deviations etc. Furthermore, to conclude that MCAK binds behind XMAP215, the authors should look at the localization of the two proteins simultaneously, on the same microtubule end. Notably, EB binding profiles are well known to exponentially decay along the microtubule lattice - this is not very apparent from the presented data. If MCAK's autonomous binding pattern matches that of EB, we should be seeing an exponentially-decaying localization for MCAK as well? However, averaged MCAK signals seem to only be fitted to Gaussian. Note that the EB binding region (i.e. position and size of the EB comet) can be substantially modulated by increasing the microtubule growth rate - this can be easily accomplished by increasing tubulin concentrations or the addition of XMAP215 (e.g. see Maurer et al. Cur Bio 2014). Thus to establish that MCAK on its own binds the same region as EB, experiments that directly modulate the size and the position of this region should be added.

(1) We thank the reviewer for this comment. Regarding the accuracy in microtubule end positioning, we now provide more details, and please see pages 18-19, lines 625-645 in the revised manuscript.

(2) Regarding the relative localization of XMAP215 and MCAK, we performed additional experiments to record their colocalizations simultaneously, on the same microtubule end. Our results showed that MCAK predominantly binds behind XMAP215, with 14.5% appearing within the XMAP215’s binding region. Please see Fig. 2.D-E and lines 184-197 in the revised manuscript.

(3) Regarding the exponential decay of the EB1 signal along microtubules, we observed that the position probability distribution measured in the present study follows a Gaussian distribution, and the expected exponential decay was not apparent. Since the exponential decay is thought to result from the time delay between tubulin polymerization and GTP hydrolysis, slower polymerization is expected to reduce this latency (Maurer et al., 2014). In our experiments, the growth rate was relatively low (~0.7 mm/min), much slower than the rate observed in cells, where the comet-shaped EB1 signal is most pronounced. The previous study has shown that the exponential decay of EB1 is more pronounced at growth rates exceeding 3 mm/min in vitro (Maurer et al., 2014). Therefore, we think that the relatively slow growth may account for the observed non-exponential decay distribution of the EB1 signals. The same reason may also explain the distribution of MCAK.

(4) We agree with the reviewer’s suggestion that altering microtubule growth rate is a valid and effective approach to regulate the EB cap length. However, the conclusion that MCAK binds to the EB region is supported by three lines of evidence: (1) the localization of MCAK at the ends of microtubules, (2) new experimental data showing that MCAK binds to the proximal end of the XMAP215 site, and (3) the tendency of MCAK to bind GTPγS microtubules, similar to EB1. Based on these findings, we did not pursue additional experiments to modify the length of the EB cap.

(2) Even if MCAK indeed binds behind XMAP215, there is no evidence that this region is defined by the GDP-Pi nucleotide state; it could still be curved protofilaments. GTPyS is an analogue of GTP - to what extent GTPyS microtubules exactly mimic the GDP-Pi-tubulin state remains controversial. Furthermore, nucleotide sensing for EB is thought to be achieved through its binding at the interface of four tubulin dimers. However MCAK's binding site is distinct, and it has been shown to recognize intradimer tubulin curvature. Thus it is not clear how MCAK would sense the nucleotide state. On the other hand, there is mounting evidence that the morphology of the growing microtubule end can be highly variable, and that curved protofilaments may be protruding off the growing ends for tens of nanometers or more, previously observed both by EM as well as by fluorescence (e.g. Mcintosh, Moores, Chretien, Odde, Gardner, Akhmanova, Hancock, Zanic labs). Thus, to establish that MCAK indeed localizes along the closed lattice, EM approaches should be used.

First, we conducted additional experiments that demonstrate MCAK indeed binds behind XMAP215, supporting the conclusion that MCAK interacts with the EB cap (please see Fig. 2 in the revised manuscript). Second, our argument that MCAK preferentially binds to GDP-Pi tubulin is based on two observations: (1) the binding regions of MCAK overlap with those of EB1, and (2) MCAK preferentially binds to GTPγS microtubules, which are considered a close analogue of GDP-Pi tubulin. Third, understanding the structural basis of how MCAK senses the nucleotide state of tubulin is beyond the scope of the present study. However, inspired by the reviewer’s suggestion, we looked into the structure of the MCAK-tubulin complex. The L2 loop of MCAK makes direct contact with the interdimer interface (Trofimova et al., 2018; Wang et al., 2017), which could provide a structural basis for recognizing the changes induced by GTP hydrolysis. While this remains a hypothesis, it is certainly a promising direction for future research. Forth, we agree with the reviewer that an EM approach would be ideal for establishing that MCAK localizes along the closed lattice. However, this is not the focus of the current study. Instead, we argue that MCAK binds to the EB cap, where at least some lateral interactions are likely to have formed.

(3) The physiological relevance of the study is rather questionable: MCAK has been previously established to be able to both diffuse along the microtubule lattice (e.g. Helenius et al.) as well as hitchhike on EBs (Gouveia et al.). Given the established localization of EBs to growing microtubule ends in cells, and apparently higher affinity of MCAK for EB vs. the microtubule end itself (although direct comparisons with the literature have not been reported here), the relevance of MCAK's autonomous binding to dynamic microtubule ends is dubious.

We thank the reviewer for raising the importance of physiological relevance. Please refer to our response to the comment No.1 of reviewer 1. Briefly, we think that the end-binding affinity of MCAK makes a significant contribution for its cellular functions. To elucidate this concept, we now use a simple model shown in Supplementary Appendix-2 (see pages 49-51, lines 1246-1316). In this model, we simplified MCAK and EB1 binding to microtubule ends by considering only these two proteins while neglecting other factors (e.g. XMAP215). Specifically, we considered two scenarios: one in which both proteins freely diffuse in the cytoplasm and another where MCAK is localized to specific cellular structures, such as the centrosome or centromere. Based on the modeling results, we argue that MCAK's functional impact at microtubule ends derives both from its intrinsic end-binding capacity and its ability to strengthen the EB1-mediated end association pathway.

(4) Finally, the study seriously lacks discussion of and comparison with the existing literature on this topic. There are major omissions in citing relevant literature, such as e.g. landmark study by Kinoshita et al. Science 2001. Several findings reported here directly contradict previous findings in the literature. Direct comparison with e.g. Gouveia et al findings, Helenius et al. findings, and others need to be included. For example, Gouveia et al reported that EB is necessary for MCAK plus-end-tracking in vitro (please see Figure 1 of their manuscript). The authors should discuss how they reconcile the differences in their findings when compared to this earlier study.

We thank the reviewer for this helpful suggestion. In the revised manuscript, we have updated the text description and included comparative discussions with other relevant studies in the Discussion section. Specifically, we added comparisons with the research on XMAP215 in page 14, lines 459-472 (Barr and Gergely, 2008; Kinoshita et al., 2001; Tournebize et al., 2000). Additionally, we have compared our findings with those of Gouveia et al. and Helenius et al. regarding MCAK's preference for binding microtubule ends in page 6, lines 145-157 and page 13, 408-441, respectively (Gouveia et al., 2010; Helenius et al., 2006).

Additional specific comments:Figure 1Gouveia et al. (Figure 1) reported that MCAK does not autonomously preferentially localize to growing tips. Specifically, Gouveia et al. found equal association rates of MCAK to both the lattice and the tip in the presence of EB3delT, an EB3 construct that does not directly interact with MCAK. How can these findings be reconciled with the results presented here?

We are uncertain why there was no observed difference in the on-rates to the lattice and the end in the study by Gouveia et al. Even when considering only the known affinity of MCAK for curved protofilaments at the distal tip of growing microtubules, we would still expect to observe an end-binding preference. After carefully comparing the experimental conditions, we nevertheless identified some differences. First, we used a 160 nm tip size to calculate the on-rate (k_on_), whereas Gouveia et al. used a 450 nm tip. Using a longer tip size would naturally lead to a smaller(k_on_) value. Note that we chose 160 nm for several reasons: (i) a previous cryo-electron tomography study has elucidated that the sheet structures of dynamic microtubule ends have an average length of around 180 nm (Guesdon et al., 2016); (ii) Analysis of fluorescence signals at dynamic microtubule ends has demonstrated that the taper length at the microtubule end is less than 180 nm (Maurer et al., 2014); (iii) in the present study, we estimated that the length of MCAK's end-binding region is approximately 160 nm. Second, in Gouveia et al., single-molecule binding events were recorded in the presence of 75 nM EB3ΔT, which could potentially create a crowded environment at the tip, reducing MCAK binding. Third, as mentioned in our response to Reviewer 1, we took great care to minimize the interference from purification tags (e.g., His-tag) by ensuring their complete removal during protein preparation. Previous studies reported that retaining the His-tag of MAPs led to a significant increase in binding for microtubules (Maurer et al., 2011; Zhu et al., 2009). We believe that some of the factors mentioned above, or their combined effects, may account for the differences in these two observations.

1C shows the decay of tubulin signal over several hundred nm - should show individual traces? How aligned? Doesn't this long decay suggest protruding protofilaments? (E.g. Odde/Gardner work).

(1) In the revised manuscript, we now show individual traces (e.g. in Fig. 1B and Fig. 2A). The average trace for tubulin signal with standard deviation was shown in Fig. 2C.

(2) The microtubule lattice was considered as a Gaussian wall and its end as a half-Gaussian in every frame. Use the peak position of the half-Gaussian of every frame to align and average microtubule end signals, during the dwell time. The average microtubule ends' half-Gaussion peak used as a reference to measure the intensity profile of individual single-molecule binding event in every frame (see page18, lines 607-624).

(3) We think that the decay of tubulin signal results from the convolution of the tapered end structure and the point spread function. In the revised manuscript, we have updated the Figures to provide unprocessed original data in Fig. 1B and Fig. 2A.

Please show absolute numbers of measurements in 1C (rather than normalized distribution only).

In the revised manuscript, we have included the raw data for both tubulin and MCAK signals as part of the methods description. In Fig. 1, using normalized values allows for the simultaneous representation of microtubule and protein signals on a unified graph.

How do the results in 1D-G compare with the previous literature? Particularly comparison of on-rates between this study and the Gouveia et al? Assuming 1 um = 1625 dimers, it appears that in the presence of EB3, the on-rate of MCAK to the tips reported in Gouveia et al. is an order of magnitude higher than reported here in the absence of EB3 (4.3 x 10E-4 vs. 2 x 10E-5). If so, and given the robust presence of EB proteins at growing microtubule ends in cells, this would invalidate the potential physiological relevance of the current study. Note that the dwell times measured in Gouveia et al. are also longer than those measured here.

Note that in Gouveia et al, the concentration of mCherry-EB3 was 75 nM, about 187.5 times higher than that of MCAK (0.4 nM). The relative concentrations of these two proteins are not always the case in cells. Regarding the physiological relevance of the end-binding affinity of MCAK itself, please refer to our response to the point No.1 of Reviewer 1.

Notably, Helenius et al reported a diffusion constant for MCAK of 0.38 um^2/s, which is more than an order of magnitude higher than reported here. The authors should comment on this!

In the revised manuscript, we have provided an explanation for the difference in diffusion coefficient. Please see page 6, line 142-157. In short, low salt condition facilitates rapid diffusion of MCAK.

Figure 2:This figure is critical and really depends on the analysis of the tubulin signal. Note significant variability in tubulin signal between presented examples in 2A. Also, while 2C looks qualitatively similar, there appears to be significant variability over the several hundred nm from the tip along the lattice. This is the crucial region; statistical significance testing should be presented. More detailed info, including SDs etc. is necessary.

In the revised manuscript, we have provided raw data in Fig. 1B and Fig. 2A. Additionally, we have provided statistical analysis on the tubulin signals (Fig. 2C) and performed significance test. Please see page 5, lines 111-116 and page 7, lines 179-183 for detailed descriptions.

Insights into the morphology of microtubule ends based on TIRF imaging have been previously gained in the literature, with reports of extended tip structures/protruding protofilaments (see e.g. Coombes et al. Cur Bio 2013, based on the methods of Demchouk et al. 2011). Such analysis should be performed here as well, if we are to conclude that nucleotide state alone, as opposed to the end morphology, specifies MCAK's tip localization.

We appreciate the reviewer’s suggestion and agree that it provides a valid optical microscopy-based approach for estimating microtubule end morphology. However, this method did not establish a direct correlation between microtubule end morphology and tubulin nucleotide status. Therefore, we think that refining the measurement of microtubule end morphology will not necessarily provide more information to the understanding of tubulin nucleotide status at MCAK binding sites. Based on the available data in the present study, there are two main pieces of evidence supporting the idea that MCAK can sense tubulin nucleotide status: (1) the binding regions of MCAK and EB overlap significantly, and (2) MCAK shows a clear preference for binding to GTPγS microtubules, similar to EB1 (we provide a new control to support this, Fig. s4). Of course, we do not consider this to be a perfect set of evidence. As the reviewer has pointed out here and in other suggestions, future work should aim to further distinguish the nucleotide status of tubulin in the dynamic versus non-dynamic regions at the ends of microtubules, and to investigate the structural basis by which MCAK recognizes tubulin nucleotide status.

EB comet profile should be clearly reproduced. MCAK should follow the comet profile.

Please see our 3^rd^ response to the point 1 of this reviewer.

The conclusion that the MCAK binding region is larger than XMAP215 is not firm, based on the data presented. The authors state that 'the binding region of MCAK was longer than that of XMAP215'. What is the exact width of the region of the XMAP215 localization and how much longer is the MCAK end-binding region? Is this statistically significant?

We have revised this part in the revised manuscript (page 6, lines 167-172). The position probability distributions of MCAK and XMAP215 were significantly different (K-S test, p< 10^-5^), and the binding region of MCAK (FWHM=185 nm) was significantly longer than that of XMAP215 (FWHM=123 nm).

MCAK localization with AMPPNP should also be performed here. Even low concentrations of MCAK have been shown to induce microtubule catastrophe/end depolymerization. This will dramatically affect microtubule end morphology, and thus apparent positioning of MCAK at the end.

In the end positioning experiment, we used a low concentration of MCAK (1 nM). Under this condition, microtubule dynamics remained unchanged, and the morphology of the microtubule ends was comparable across different conditions (with EB1, MCAK or XMAP215). Additionally, in the revised manuscript, we present a new experiment in which we recorded the localization of both MCAK and XMAP215 on the same microtubule. The results support the conclusion regarding their relative localization: most MCAK is found at the proximal end of the XMAP215 binding region, while approximately 15% of MCAK is located within the XMAP215 binding region. Please see Fig. 2D-E and page 7, lines 184-197 for the corresponding descriptions.

Figure 3:For clearer presentation, projections showing two microtubule lattice types on the same image (in e.g. two different colors) should be shown first without MCAK, and then with MCAK.

We thank the reviewer for this suggestion. We have adjusted the figure accordingly. Please see Fig. 4 in the revised manuscript.

Please comment on absolute intensity values - scales seem to be incredibly variable.

The fluorescence value presented here is the result of multiple images being summed. Therefore, the difference in absolute values is influenced not only by the binding affinity of MCAK in different states to microtubules, but also by the number of images used. In this analysis, we are not comparing MCAK in different states, but rather evaluating the binding ability of MCAK in the same state on different types of microtubules.

Given that the authors conclude that MCAK binding mimics that of EB, EB intensity measurements and ratios on different lattice substrates should be performed as a positive control.

We performed additional experiments with EB1, in the revised manuscript, we provide the data as a positive control (please see Fig. s4).

Figure 4:MCAK-nucleotide dependence of GMPCPP microtubule-end binding has been previously established (see e.g. Helenius et al, others?) - what is new here? Need to discuss the literature. This would be more appropriate as a supplemental figure?

In the present study, we reproduced the GMPCPP microtubule-end binding of MCAK in the AMPPNP state, as shown in several previous reports (Desai et al., 1999; Hertzer et al., 2006). Here, we also quantified the end to lattice binding preference, and our results showed that the nucleotide state-dependence shows the same trend as the binding preference of MCAK to the growing microtubule ends. Therefore, we prefer to keep this figure in the main text (Fig. 5).

Figure 5:Please note that both MCAK mutants show an additional two orders of magnitude lower microtubule binding on-rates when compared to wt MCAK. This makes the analysis of preferential binding substrate for these mutants dubious.

We agreed with this point. We have rewritten this part. Please see page 10, lines 295-327, in the revised manuscript.

Figure 6:Combined effects of XMAP215 and XKCM1 (MCAK) have been previously explored in the landmark study by Kinoshita et al. Science 2001, which should be cited and discussed. Also note that Moriwaki et al. JCB 2016 explored the combined effects of XMA215 and MCAK - which should be discussed here and compared to the current results.

We agree with the reviewer. We have revised the discussion on this part. Please see page 11, lines 329-342 and page 14, lines 459-472 in the revised manuscript.

Please report quantification for growth rate and lifetime.

In the revised manuscript, we provide all these data. Please see pages 11-12, lines 343-374.

To obtain any new quantitative information on the combined effects of the two proteins, at the very minimum, the authors should perform a titration in protein concentration.

We agree with the reviewer on this point. In our pilot experiments, we performed titration experiments to determine the appropriate concentrations of MCAK and XMAP215, respectively. We selected 50 nM for XMAP215, as it clearly enhances the growth rate and exhibits a mild promoting effect on catastrophe—two key effects of XMAP215 reported in previous studies (Brouhard et al., 2008; Farmer et al., 2021). Reducing the XMAP215 concentration eliminates the catastrophe-promoting effect, while increasing it would not much enhance the growth rate. For MCAK, we chose 20 nM, as it effectively promotes catastrophe; increasing the concentration beyond this point leads to no microtubule growth, at least in the MCAK-only condition. If there’s no microtubule growth, it would be difficult to quantify the parameters of microtubule dynamics, hindering a clear comparison of the combined versus individual effects. Therefore, we think that the concentrations used in this study are appropriate and representative. In the revised manuscript, we make this point clearer (see pages 11 and lines 329-342).

Finally, the writing could be improved for overall clarity.

We thank the reviewer for pointing out this. In the revised manuscript, we conducted a thorough revision and review of the text.

**Reviewer #3 (Public Review):**
The authors revisit an old question of how MCAK goes to microtubule ends, partially answered by many groups over the years. The authors seem to have omitted the literature on MCAK in the past 10-15 years. The novelty is limited due to what has previously been done on the question. Previous work showed MCAK targets to microtubule plus-ends in cells through association with EB proteins and Kif18b (work from Wordeman, Medema, Walczak, Welburn, Akhmanova) but none of their work is cited.

We thank the reviewer for the suggestion. Some of the referenced work has already been cited in our manuscript, such as studies on the interaction between MCAK and EB1. However, other relevant literature had not been properly cited. In the revised manuscript, we have added further discussion on this topic in the context of existing findings. Please refer to pages 3-4, lines 68-85, and pages 13, lines 425-441.

It is not obvious in the paper that these in vitro studies only reveal microtubule end targeting, rather than plus end targeting. MCAK diffuses on the lattice to both ends and its conformation and association with the lattice and ends has also been addressed by other groups-not cited here. I want to particularly highlight the work from Friel's lab where they identified a CDK phosphomimetic mutant close to helix4 which reduces the end preference of MCAK. This residue is very close to the one mutated in this study and is highly relevant because it is a site that is phosphorylated in vivo. This study and the mutant produced here suggest a charge-based recognition of the end of microtubules.Here the authors analyze this MCAK recognition of the lattice and microtubule ends, with different nucleotide states of MCAK and in the presence of different nucleotide states for the microtubule lattice. The main conclusion is that MCAK affinity for microtubules varies in the presence of different nucleotides (ATP and analogs) which was partially known already. How different nucleotide states of the microtubule lattice influence MCAK binding is novel. This information will be interesting to researchers working on the mechanism of motors and microtubules. However, there are some issues with some experiments. In the paper, the authors say they measure MCAK residency of growing end microtubules, but in the kymographs, the microtubules don't appear dynamic - in addition, in Figure 1A, MCAK is at microtubule ends and does not cause depolymerization. I would have expected to see depolymerization of the microtubule after MCAK targeting. The MCAK mutants are not well characterized. Do they still have ATPase activity? Are they folded? Can the authors also highlight T537 and discuss this?Finally, a few experiments are done with MCAK and XMAP215, after the authors say they have demonstrated the binding sites overlap. The data supporting this statement were not obvious and the conclusions that the effect of the two molecules are additive would argue against competing binding sites. Overall, while there are some interesting quantitative measurements of MCAK on microtubules - in particular in relation to the nucleotide state of the microtubule lattice - the insights into end-recognition are modest and do not address or discuss how it might happen in cells. Often the number of events is not recorded. Histograms with large SEM bars are presented, so it is hard to get a good idea of data distribution and robustness. Figures lack annotations. This compromises therefore their quantifications and conclusions. The discussion was hard to follow and needs streamlining, as well as putting their work in the context of what is known from other groups who produced work on this in the past few years.

We thank the reviewer for the comments. Regarding the physiological relevance of the end-binding of MCAK itself, please refer to our response to the point No.1 of reviewer 1. Moreover, as we feel that other suggestions are more thoroughly expressed in the following comments for authors, we will provide the responses in the corresponding sections, as shown below.

**Reviewer #3 (Recommendations For The Authors):**
Why, on dynamic microtubules, is MCAK at microtubule plus ends and does not cause a catastrophe?

At this concentration (10 nM MCAK with 16 mM tubulin in Fig. 1; 1 nM MCAK with 12 mM tubulin in Fig. 2), MCAK has little effect on microtubule dynamics in our experiments. Using TIRFM, we were able to observe individual MCAK binding events. Based on these observations, we think that in the current experimental condition, a single binding event of MCAK is insufficient to induce microtubule catastrophe; rather, it likely requires cumulative changes resulting from multiple binding events.

Do the MCAK mutants still have ATPase activity?

The ATPase activities of MCAK^K525A^ and MCAK^V298S^ are both reduced to about 1/3 of the wild-type (Fig. s6).

The intensities of GFP are not all the same on the microtubule lattice (eg 1A). See blue and white arrowheads. The authors could be looking at multiple molecules of GFP-MCAK instead of single dimers. How do they account for this possibility?

In the revised manuscript, we provide the gel filtration result of the purified MCAK, and the position of the peak corresponds to ~220 kDa, demonstrating that the purified MCAK in solution is dimeric (please see Fig.s1 and page 5, lines 101-103). We measured the fluorescence intensity of each binding event. A probability distribution of these intensities was then constructed and fitted with a Gaussian function. A binding event was considered to correspond to a single molecule if its intensity fell within μ±2σ of the distribution. The details of the single-molecule screening process are provided in the revised manuscript (see page 17, lines 574-583).

In addition, we also measured the fluorescence intensity of both MCAK^sN+M^ and MCAK. MCAK^sN+M^ is a monomeric mutant that contains the neck domain and motor domain (Wang et al., 2012). The average intensity of MCAK^sN+M^ is 196 A.U., about 65 % of that of MCAK (300 A.U.), suggesting that MCAK is a dimer (see Fig. s1). Moreover, we think that some of the dim signals may result from stochastic background noise, while others likely represent transient bindings of MCAK. The exposure time in our experiments was approximately 0.05 seconds; if the binding duration were shorter than this, the signal would be lower. It is important to note that in this study, we specifically selected binding events lasting at least 2 consecutive frames, meaning transient binding events were not included. This point has been clarified in the Methods section (see page 17, lines 568-569 and lines 574-583).

Could the authors provide a kymograph of an MT growing, in the presence of MCAK+AMPPNP? Can MCAK track the cap?

Under single-molecule conditions, we observed a single MCAK molecule briefly binding to the end of the microtubule. However, we did not record if MCAK at high concentrations could track microtubule ends under AMPPNP conditions.

In the experiments in Figure 6, the authors should also show the localization of MCAK and XMAP215 at microtubule plus ends in their kymographs to show the two molecules overlap.

Regarding the relative localization of XMAP215 and MCAK, we conducted additional experiments to record their colocalization simultaneously at the same microtubule end. Our results show that MCAK predominantly binds behind XMAP215, with 14.5% of MCAK binding within the XMAP215 binding region. Please see Fig. 2.D-E and page 7, lines 184-197 in the revised manuscript. However, we argue that the effects of XMAP215 and MCAK are additive, and their binding sites do not necessarily need to overlap for these effects to occur.

The authors do not report what statistical tests are done in their graphs, and one concern is over error propagation of their data. Instead of bar graphs, showing the data points would be helpful.

We have now shown all data points in the revised manuscript.

MCAK+AMPPNP accumulates at microtubule ends. Appropriate quotes from previous work should be provided.

We have made the revisions accordingly. Please see page 9, lines 273-276.

Controls are missing. An SEC profile for all purified proteins should be presented. Also, the authors need to explain if they report the dimeric or monomeric concentration of MCAK, XMAP215, etc...

We have provided the gel filtration result for all purified proteins in the revised manuscript (Fig.s1). Moreover, we now make it clear that the concentrations of MCAK and EB1 are monomeric concentration. Please see the legend for Fig. 1, line 893 in the revised manuscript.

Figure 1: the microtubules don't look dynamic at all. This is also why the authors can record MCAK at microtubule ends, because their structure is not changing.

The microtubules are dynamic, but they may appear non-dynamic due to the relatively slow growth rate and the high frame rate at which we are recording. We propose that individual binding events of MCAK induce structural changes at the nanoscopic or molecular scale, which are not detectable using TIRFM.

I recommend the authors measure the Kon and Koff for single GFP-MCAK mutant molecules and provide the information alongside their normalized and averaged binding intensities of GFP-MCAK in Fig 5. Showing data points instead of bar graphs would be better.

(1) We measured k_on_ and dwell time for mutants at growing microtubule end. However, we did not perform single-molecule tracking for MCAK’s binding on stabilized microtubules. This is mainly because the superimposed signal on the stable microtubule already indicates the changes in the mutant's binding affinity to different microtubule structures, and moreover, the binding of the mutants is highly transient, making accurate single-molecule tracking and calculations difficult.

(2) In the revised figure, we have included the data points in all plots.

When discussing how Kinesin-13 interacts with the lattice, the authors should quote the papers that report the organization of full-length Kinesin-13 on tubulin heterodimers: Trofimova et al, 2018; McHugh et al 2019; Benoit et al, 2018. It would reinforce their model and account for the full-length protein, rather than just the motor domain.

We thank the suggestion for the reviewer. In our manuscript, we have cited papers on full-length Kinesin-13 to discuss the interaction between MCAK and microtubule end-curved structure. Additionally, we have utilized the MCAK-tubulin crystal structure (PDB ID: 5MIO) in Fig. 6, as it depicts a human MCAK, which is consistent with the protein used in our study. This structure illustrates the interaction sites between MCAK and tubulin dimer, guiding our mutation studies on specific residues. Thus, we prefer to use the structure (PDB ID: 5MIO) in Fig.6.

Figure 5A. What type of model is this? A PDB code is mentioned. Is this from an X-ray structure? If so, mention it.

We have now included the structural information in the Figure legend (see page 37, lines 1045).

Figure 5B. It is not possible to distinguish the different microtubule lattices (GTPyS, GDP, and GMPCPP). The experiment needs to be better labelled.

We thank the reviewer for this comment. We have now rearranged the figure for better clarity (see Fig. 6).

"Figure 5D: what are the statistical tests? I don't understand " The statistical comparisons were made versus the corresponding value of 848 GFP-MCAK".

We have made this point clearer in the revised manuscript (see pages 38, line 1078-1080).

What is the "EB cap"? This needs explaining.

We provide this explanation for this, please see page 4, lines 87-89 in the revised manuscript.

Work from Friel and co-workers showed MCAK T537E did not have depolymerizing activity and a reduced affinity for microtubule ends. The work of the authors should be discussed with respect to this previously published work.

We thank the reviewer for this suggestion. In the revised manuscript, we have added discussions on this (see page 10, lines 303-307).

The concentration of protein used in the assays is not always described.

We have checked throughout the manuscript and made revisions accordingly.

"Having revealed the novel binding sites of MCAK in dynamic microtubule ends " should be on "we wondered how MCAK may work ..with EB1". This is not addressed so should be removed. Instead, they can quote the work from Akhmanova's lab. Realistically this section should be rephrased as there are other plus-end targeting molecules that compete with MCAK, not just XMAP215 and EB1.

We have rephrased this section as suggested by this reviewer to be more specific. Please see page 11, lines 329-342.

What is AMPCPP?

It should be “AMPPNP”

Typos in Figure 5.

Corrected